# Composition of nasal bacterial community and its seasonal variation in health care workers stationed in a clinical research laboratory

Nazima Habibi[1¤a]*, Abu Salim Mustafa[1,2]*, Mohd Wasif Khan[1¤b]

1 OMICS Research Unit and Research Core Facility, Faculty of Medicine, Health Sciences Centre, Kuwait University, Jabriya, Kuwait, 2 Department of Microbiology, Faculty of Medicine, Health Sciences Centre, Kuwait University, Jabriya, Kuwait

¤a Current address: Biotechnology Program, Environment and Life Science Research Centre, Kuwait Institute for Scientific Research, Kuwait City, Kuwait
¤b Current address: Department of Biochemistry and Medical Genetics, University of Manitoba, Winnipeg, MB, Canada
* nhabibi@kisr.edu.kw (NH); abusalim@hsc.edu.kw (ASM)

**Data Availability Statement:** The data can be accessed through these URLs: https://www.mg-rast.org/linkin.cgi?project=mgp16453; https://www.mg-rast.org/linkin.cgi?project=mgp16510;

## Abstract

The microorganisms at the workplace contribute towards a large portion of the biodiversity a person encounters in his or her life. Health care professionals are often at risk due to their frontline nature of work. Competition and cooperation between nasal bacterial communities of individuals working in a health care setting have been shown to mediate pathogenic microbes. Therefore, we investigated the nasal bacterial community of 47 healthy individuals working in a clinical research laboratory in Kuwait. The taxonomic profiling and core microbiome analysis identified three pre-dominant genera as *Corynebacterium* (15.0%), *Staphylococcus* (10.3%) and, *Moraxella* (10.0%). All the bacterial genera exhibited seasonal variations in summer, winter, autumn and spring. SparCC correlation network analysis revealed positive and negative correlations among the classified genera. A rich set of 16 genera (q < 0.05) were significantly differentially abundant (LEfSe) across the four seasons. The highest species counts, richness and evenness (P < 0.005) were recorded in autumn. Community structure profiling indicated that the entire bacterial population followed a seasonal distribution ($R^2$-0.371; P < 0.001). Other demographic factors such as age, gender and, ethnicity contributed minimally towards community clustering in a closed indoor laboratory setting. Intra-personal diversity also witnessed rich species variety (maximum 6.8 folds). Seasonal changes in the indoor working place in conjunction with the outdoor atmosphere seems to be important for the variations in the nasal bacterial communities of professionals working in a health care setting.

https://www.mg-rast.org/linkin.cgi?project=mgp16544; https://www.mg-rast.org/linkin.cgi?project=mgp16599; https://www.mg-rast.org/linkin.cgi?project=mgp16600.

**Funding:** The study was funded by Kuwait University Research Sector's grants SRUL02/13.

**Competing interests:** The authors have declared that no competing interests exist.

## Introduction

The human body houses 10 to 100 trillion diverse bacteria, which are almost equal to the cells in our body [1]. Each site in a human body act as a distinct ecosystem, be it the gut, mouth, scalp, skin, and, any other crevices or orifices. The roles of microbes present in these unique habitats span from pathogenic, symbiotic to harmless forms [2, 3]. The nose is also an important site of microbial colonization [4–7]. The outermost segment of the nose, the nostrils or, anterior nares, is a transition zone from the outer environment to the windpipe and the respiratory organs [8]. It is known that opportunistic pathogens in the anterior nasal cavity spread to other sections of the respiratory tract and are involved in the development of respiratory disorders such as allergic rhinitis, chronic rhinosinusitis, asthma, pneumonia, otitis media, etc. [9].

The microorganisms at the workplace contribute towards a large portion of the biodiversity a person encounters in his or her life. Our professional associates impact our health, both positively and negatively. Dispersal of microbial communities between humans and the surrounding environment either through touching or airborne release is a key medium to transmit pathogens [10–14]. Understanding the processes that structures an individuals' microbial communities is an important endeavor since we spend the bulk of time with our colleagues. In recent years, the scientific community has begun to recognize the significance of characterizing such human-associated habitats, with an increasing number of studies seeking to determine the biodiversity, ecology and, public health implications of microbial assemblages present therein [15].

Professionals in hospitals and associated clinical laboratories are at increased risk of cross infections due to the front-line nature of their work. Laboratory professionals are directly or indirectly exposed to a diverse range of pathogenic organisms. Inadvertently they become mediators of several opportunistic and multidrug-resistant pathogens [16]. The nasal cavity being an open point of entry and air exchange disperses the microbial communities into the surrounding environment as unseen aerosols [8, 15]. Bacterial outbreaks have been reported in a long term health care facility in Taiwan [17]. In a prospective study, a high incidence of human-related microorganisms of the nasal carriage was discovered at a long-term health care facility [18–20].

Competition and cooperation between nare-associated communities in a particular environment have been shown to impact the prevalence of pathogenic bacterial colonization and subsequent infection [21, 22]. Chronic respiratory diseases such as asthma and allergic rhinitis (AR) are major public health problems in developing countries including those in the Middle East. In a recent study, the allergic rhinitis burden was revealed to be significantly high in the adults of five Middle Eastern countries of Egypt, Turkey, Kuwait, Saudi Arabia and, the United Arab Emirates [23]. Allergic rhinitis and other respiratory disorders result in a negative impact on the quality of life, quality of sleep and, daily activities.

From all the above it appears essential to expand our knowledge of human nasal bacterial communities and their diversity in health care settings. The 16S rRNA gene sequencing approach has been used successfully in the detection and identification of bacteria [24–26]. With the advent of next-generation sequencing, many non-cultivable bacteria in human-associated habitats have been reported [2, 3, 27, 28]. In this study, we used a 16S rRNA amplicon sequencing to study the nasal bacterial composition of the staff of a clinical research laboratory. Four main aspects were considered for analysis and discussion in this investigation: (1) understanding the composition of the nasal microbiota of individuals working in the facility (2) identifying the core bacterial genera (3) analyzing the effect of seasonal variations on the nasal bacterial community and (4) Intra-individual diversity at different time points.

## Materials and methods

### Study subjects

Forty-seven volunteers were enrolled for the present study. Each volunteer was sampled at least once or more to collect 73 samples over one year. All of them were adults (falling into two age groups, *i.e.* A: 18–30 years, and B: 31–60 years of age) of both sexes (males and females) and two ethnicities (Arabs and non-Arabs). All the volunteers were the staff of the OMICS Research Unit/ Research Core Facility (OMICSRU), Health Sciences Center, Kuwait University, Kuwait, or the staff from other departments in the same centre. The OMICSRU (29.327679, 48.032603) is a central laboratory adjacent to the Faculty of Medicine, Health Science Centre, Kuwait University, Kuwait. It's one storey, centrally air-conditioned building, providing bench space and equipment usage facilities to the staff of the Health Science Centre. Samples were collected from the volunteers for 1 year (in April, June, November, December and February). Each month was representative of four seasons *i.e.* summer (April-June), autumn (November), winter (December) and spring (February) in Kuwait during the year 2014–2015. The temperature and the weather of each month of sample collection were recorded according to Kuwait Meteorological Department (http://www.met.gov.kw/Forecasts/kuwait.php) and given in Table 1. Ten individuals among the 47 volunteers were picked to study the intra-personal diversity (Table 2). These individuals were consecutively sampled at three different time points (November, December and February). The study was approved by the Ethics Committee of Health Sciences Centre, Kuwait University. Prior written consent was obtained from all participants. They were queried for intake of antibiotics and were excluded from the study if responded positively. The dataset from 88 professionals working in health care centres (HCC) in Taiwan [15] were used from a geographically distant health care setting to compare our data. Another dataset from 20 healthy individuals (NHC) working in a livestock farm in Iowa were used as a control to compare our dataset with healthy individuals working in non-healthcare settings [29]. These 20 individuals were not in contact with animals on the farm and worked indoors.

### Nasal swab collection and DNA isolation

A nasal specimen from each volunteer was obtained [6] from both nostrils (right and left) using a single disposable dry swab (BD CultureSwab™, France) and dipped into 2 ml sterile phosphate-buffered saline (PBS) pH-7.0 (1X, Gibco®, Life Technologies, Auckland, NZ). To avoid any fungal contamination, 0.2 μg/ml Fungizone (Gibco®, Life Technologies, Auckland, NZ) was added to the PBS. The top of the swab was aseptically snipped off into a sterile 2 ml Eppendorf tube and the PBS was also poured carefully. DNA was isolated by the QiaAmp DNA Mini Kit (Qiagen, Valencia, CA) as per the manufacturer's instructions and quantified fluorometrically through the Qubit dsDNA HS Assay system (Life Technologies, Carlsbad, CA). DNA in all the samples were normalized to 5 ng/μl for subsequent 16S amplicon sequencing [24, 25]. A plain swab without any sample was simultaneously processed for DNA isolation as a negative control. The readings in the Qubit fluorometer read as 'too low' for the negative control indicating the absence of DNA. The isolated DNA samples were stored at −80˚C until further use.

### Amplicon PCR of 16S ribosomal gene

The V3 and V4 regions of the 16S rRNA gene (~600 bp) were amplified by the recommended Forward–(S-D-Bact-0341-b-S-17): 5'–TCGTCGGCAGCGTCAGATGTGTATAAGAGACAGCC TACGGGNGGCWGCAG–3' and Reverse (S-D-Bact-0785-a-A-21): 5– 'GTCTCGTGGGCTCG

**Table 1. Sampling schedule and demographic features of health care workers stationed in a clinical research laboratory of Kuwait.**

| Season | Temperature | Weather | Total No. of individuals sampled in each season | No of the individuals sampled | | | | | |
|--------|-------------|---------|------------------------------------------------|------|--------|-------|-----------|-----------|-----------|
| | | | | Male | Female | Arabs | Non-Arabs | 18-30y (A) | 30-60y (B) |
| Summer | 46 ± 2.0˚C | Sunny | 20 | 7 | 13 | 5 | 15 | 8 | 12 |
| Autumn | 25 ± 3.0˚C | Cloudy | 14 | 5 | 9 | 2 | 12 | 6 | 8 |
| Winter | 20 ± 1.0˚C | Sunny | 17 | 8 | 9 | 3 | 14 | 11 | 6 |
| Spring | 16 ± 1.5˚C | Dusty & Cloudy | 22 | 10 | 12 | 7 | 15 | 13 | 9 |
| | | Total | 73 | 30 | 43 | 17 | 56 | 38 | 35 |
| | | Grand Total | | 73 | | 73 | | 73 | |

Temperature average is recorded for a particular month. The coordinates of the sample collection site are 29.3276˚ N, 48.0333˚ E

GAGATGTGTATAAGAGACAGGACTACHVGGGTATCTAATC–3') primer pairs [30, 31]. Both primers were synthesized commercially and purchased from Thermo Fisher Scientific (Waltham, MA, USA), dissolved in 0.1 mM Tris EDTA and adjusted to 1 μM concentration. The PCR reaction mixtures (25 μl) contained 2.5 μl of sample DNA (5ng/ μl), 12.5 μl of 2x KAPA HiFi HotStart Ready Mix (Kapa Biosystems, Boston, MA) and 5 μl each forward and reverse primer. Standard positive (E. coli-1 ng/μl) and negative controls (Nuclease free water) were used in all PCR reactions. The PCR was carried out in the GeneAmp® PCR System 9700 (Applied Biosystems, Grand Island, NY) with initial activation of the DNA polymerase at 95˚C for 3 min, followed by 35 cycles at 95˚C for 30 s, 55˚C for 30 s and 72˚C for 30 s, and a final extension step at 72˚C for 5 min [27]. The PCR products were visualized on a Bioanalyzer DNA1000 chip (Agilent 2100, Santa Clara, CA) and showed a trace of ~ 550 bp (S1 Fig in S1 Appendix). Post-PCR clean-up was done by the Agencourt AMPure XP magnetic beads (Beckman Coulter Genomics, Miami, FL), according to the manufacturer's instructions.

## Index PCR, normalization and sequencing

The purified PCR products were further processed for indexing PCR by the Nextera XT Index kit (Illumina, San Diego, CA). In brief, a reaction mixture of 50 μl was prepared by adding 25 μl of 2x KAPA HiFi HotStart Ready-mix (Kapa Biosystems), 5 μl each of the Index 1 and Index 2 primers, and 10 μl of Nuclease free water (Millipore) to 5 μl of the purified PCR-product. The PCR conditions were initial activation at 95˚C for 3 min, thereafter 8 cycles of

**Table 2. Working area of ten health care professionals sampled at three different time points.**

| Sample Id | Floor | Laboratory Name/Number | Working Space |
|-----------|-------|------------------------|---------------|
| P01 | FF | First Lab | Immunology area |
| P02 | GF | Genomics Lab 1 | PCR and Gel Doc bench |
| P03 | FF | Fourth Lab | Tissue Culture area |
| P04 | GF | Proteomics Lab | NGS bench |
| P05 | GF | Genomics Lab2 | DNA/RNA isolation area |
| P06 | GF | Genomics Lab 1 | Near the main office |
| P07 | GF | Genomics Lab 1 | DNA Sequencing bench |
| P08 | FF | Third Lab | Flow Cytometry lab |
| P09 | GF | Genomics Lab 1 | Bioinformatics workstation |
| P10 | FF | Second Lab | Microscopy lab |

FF-First floor; GF-Ground floor

denaturation at 95˚C for 30 s, annealing at 55˚C for 30 s and extension at 72˚C for 30 s followed by a final extension at 72˚C for 30s. A PCR cleaning step was performed using the Agencourt AMPure XP magnetic beads (Beckman Coulter Genomics), as mentioned in the Illumina protocol. The band size of amplified DNA was verified by loading a 1:50 dilution of the final library on a DNA 1000 chip in the Bioanalyzer (Agilent). All the products gave a clear peak at ~ 630 ± 5 bp. The libraries were quantified fluorometrically on a Qubit fluorometer (Invitrogen, Carlsbad, CA). All the libraries were normalized at 4 nM and 5 μl of an aliquot of each sample was pooled together. The samples were loaded on a MiSeq v2 500 cycle cartridge (Illumina, San Diego, CA) and sequenced according to the metagenomics workflow. Raw sequences (fastq format) were subjected to the FastQC v 0.11.9 to check the basic statistics and average quality values [32]. The files were subsequently uploaded to the online server of Metagenomic Rapid Annotations using Subsystems Technology (MG-RAST) version 3.6 [33, 34] (S1 Table in S1 Appendix). All the data is publicly available on the MG-RAST platform and can be accessed through these links https://www.mg-rast.org/linkin.cgi?project=mgp16453; https://www.mg-rast.org/linkin.cgi?project=mgp16510; https://www.mg-rast.org/linkin.cgi?project=mgp16544; https://www.mg-rast.org/linkin.cgi?project=mgp16599; https://www.mg-rast.org/linkin.cgi?project=mgp16600.

### Bioinformatics and statistical analysis

Standard bioinformatics pipelines were used for downstream analysis and interpretations. Low-quality reads were filtered via SolexaQA [35]. Host contamination removal was done in Bowtie [36]. The BLAT [37] similarity search against the M5rna database [38] @ 97% cut-off revealed a total of 2363 OTUs within the dataset. Subsequent analysis and visualization were executed on the online MicrobiomeAnalyst server [39]. Taxonomic profiling was performed on rarified data with a Good's coverage of *ca*. 98% for all the samples. Mean, the maximum and minimum percentage of prevalent genera were estimated in R software [40]. Total sum scaling (TSS) was applied for core microbiome analysis [41]. The core microbiome was defined as genera present in 30% of the samples [42]. Differential tree analysis (Wilcoxon Rank Sum p < 0.05) was conducted on data normalized through centered log-ratio (CLR) [43]. The SparCC algorithm (p < 0.05) was employed for correlation pattern search and correlation network analysis [44] at a threshold of 0.3 on 100 permutations. Six alpha diversity parameters (Observed, Chao1, ACE, Shannon, Simpson, and Fisher) were compared for seasonality. All comparisons were done by the Student's t-test or ANOVA (Analysis of Variance) at a confidence interval of 95% [41] and pairwise differential abundance analysis [43]. Paired statistical analysis were also performed. For community profiling, beta diversity-based analysis of similarities (ANOSIM) on BRAY Curtis distances was applied on data normalized through total sum scaling (TSS). Results were plotted on a principal coordinate analysis (PCoA) graph. Hierarchical clustering and dendrogram analysis were performed on the Euclidean and Bray Curtis distances respectively, through the WARD algorithm. The LEfSe procedure was used for the linear discriminant analysis (LDA) at an adjusted false discovery rate (FDR/q-value < 0.05) on OTUs normalized through relative log expression (RLE) [45].

### Results

We aimed to understand the composition of bacterial communities of the health care workers stationed in the research core facility of Kuwait. The 16S amplicon sequencing successfully provided this information. Sequencing of 73 samples yielded 6,029,453,568 sequences (average 82,595,254 sequences per sample) of 22,487,573 base pairs (bp) (average 308049 bp per sample) (S2 Fig in S1 Appendix). Approximately 10% of sequences with low quality were removed and

90% were used for taxonomic profiling. Sequences < 5 bases and quality score < 20, were filtered out. Data rarefication (S3 Fig in S1 Appendix) yielded 2363 OTUs (HC) that were used for comparison with a dataset of two groups [15, 29] *i.e.* nasal microbial communities of healthcare workers (HCC) of a health institute and non-healthcare workers (NHC) from a farm. Taxonomic profiling with both the data set revealed some common and variable genera (S2 Table in S1 Appendix). The common genera might be due to the similar ecological niche the bacterial communities shared. Comparison by LEfSe identified 31 genera to be differentially abundant among the three groups. Beta diversity clustering separated the groups as three distinct clusters (Axis 1–28%; Axis 2–12.9%) (S4 Fig in S1 Appendix). Difference in taxonomic profiles at the genus level between the HC and HCC are most likely due to the wide differences in the topographical and meteorological conditions. The variations between HC and NHC are attributed to the variation in the work place and the prevailing environmental conditions. A separate analysis of taxonomic profiling and core microbiome were conducted on the dataset obtained in the present study in order to study the bacterial community composition and seasonal variations.

## Taxonomic distribution and core microbiome

We performed the taxonomic profiling and estimated the zero-inflated microbial composition also termed as relative abundance (RA) at phylum, class, order, family, and genus level (S3 Table in S1 Appendix). Approximately 37% of the bacteria remained unclassified. The classified taxa were represented by four Phyla, *i.e.*, Proteobacteria (26%), Actinobacteria (20%), Firmicutes (13%) and Bacteriodetes (4%). These phyla were further divided into classes (n = 15), orders (n = 26) and families (n = 29). The main classes were Actinobacteria (20%), Gammaproteobacteria (18%), Bacilli (11%), Alphaproteobacteria (8%), Flavobacteria (2%), and Clostridales (2%). The major orders were Actinomycetales (20%), Pseudomonadales (15%), and Bacillales (10%). At the family level Corynebacteriaceae (19%), Moraxallaceae (13%), and Staphylococcaceae (10%) were found at the first three positions. The taxonomic classification at the genus level returned 83 classified genera. The dominant genera (RA > 1%) were *Corynebacterium* (RA-14.6%) > *Moraxella* & *Staphylococcus* (RA-10.3%) > *Pseudomonas* (RA-2.4%) > *Cytophaga* (RA-1.6%), > *Flavobacterium* (RA-1.2%) > *Myroides* (RA ~ 1%). The genera (n = 75) with RA < 1% were grouped as Others. Statistical evidence of each taxonomic level was obtained through the Wilcoxon rank test (P < 0.05) on the median abundances. The twenty most common genera and their trailing taxonomies were represented on a differential heat tree (Fig 1). A core microbiome analysis was also performed to check the prevalence (presence in the number of samples) of these 83 genera in the nasal cavity of individuals sampled in the current investigation (Fig 2). Maximum prevalence was shown by *Corynebacterium* (0.89) followed by *Staphylococcus* (0.75), *Pseudomonas* (0.56), *Moraxella* (0.44), *Myroides* (0.38) and *Flavobacterium* (0.33). The rest had prevalence below 0.30 (S4 Table in S1 Appendix).

Taxonomic profiling of the OTUs (2363) in the standard package of R returned 114 classified genera. The abundances (mean, maximum, minimum) and prevalence of these genera were estimated. The taxonomies of the top eight genera with a prevalence above 50% and Mean distribution (MD) ≥ 0.5 are presented in Table 3. These genera in ascending order of their MD were *Staphylococcus* > *Corynebacterium* > *Pseudomonas* > *Bacillus* > *Moraxella* > *Propionibacterium* > *Flavobacterium* > *Micrococcus*. Our results were in agreement with the taxonomies mentioned previously. All these genera formed the part of the core microbiome as well. Through all the above analysis we conclude that *Corynebacterium*, *Staphylococcus*, and *Moraxella* were the most stable and predominant bacterial genera of the nasal cavities of health care professionals working at the research core facility of Kuwait during the sampling period.

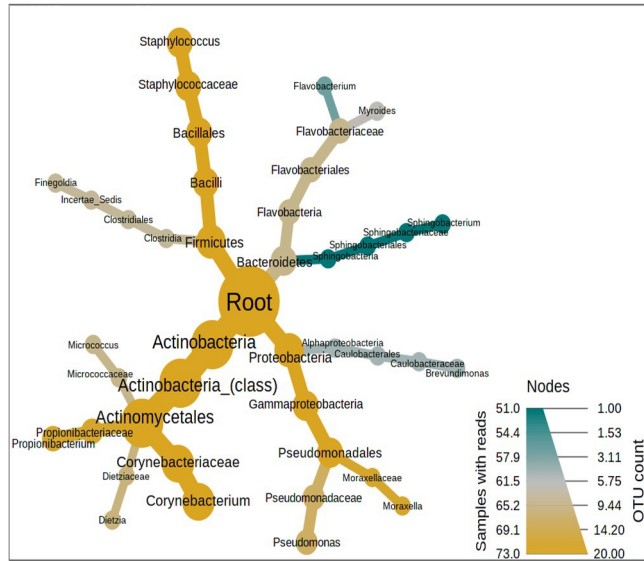

**Fig 1. Taxonomic distribution of nasal bacterial population represented on a differential heat tree.** The most common OTUs were picked by setting the parameters of minimum count as 10, sample prevalence as 50% and inter quartile range of 20%. The nodes in yellow represent the highly abundant taxa. The Wilcoxon rank test (P < 0.05) was applied on the median abundances of the chosen 20 OTUs.

## Seasonal variations

We presumed the genera distribution to be affected due to the temporal changes in the working atmosphere. This was demonstrated by the LDA analysis revealing 16 bacterial genera to be significantly differentially abundant across the four seasons (Fig 3). High LDA scores in Autumn were recorded for *Ensifer*, *Myroides*, *Propionibacterium*, *Bosea*, and *Micrococcus*. Only *Flavobacterium* was lavishly abundant in spring. Key genera with extraordinary LDA scores in summer were *Bradyrhizobium*, *Phyllobacterium*, and *Kocuria*. In winter, among the enriched genera were *Cytophaga*, *Pseudomonas*, *Agrobacterium*, *Stenotrophomonas*, *Brevundimonas*, and *Rhodococcus*.

We observed the constant presence of prevalent genera in all four seasons but, their RA varied considerably. In autumn *Staphylococcus* was dominant (RA 23.2%). Very close to its RA was *Corynebacterium* (RA-21.5%) followed by *Moraxella* (RA- 15%). In spring *Corynebacterium* (RA- 12.3%) was on the top followed by *Moraxella* (RA-10%) and *Staphylococcus* (7.0%). In summer *Corynebacterium* and *Moraxella* (RA-17%) were followed by *Staphylococcus* (RA-7.0%). *Corynebacterium* continued to exhibit the highest RA (30%) in autumn as well and was

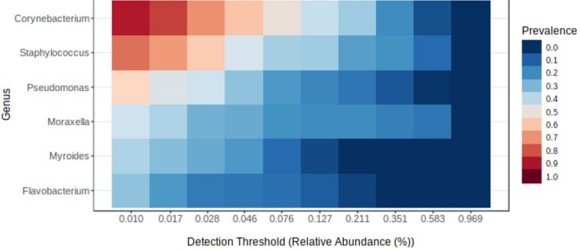

**Fig 2. Core bacteriome of nasal samples of health care professionals stationed in a clinical research laboratory.** A set of taxa detected in 30% of the population with a prevalence above 0.01 were plotted on the heat map. The relative abundance derived from the count data is plotted on the X-axis. The Y-axis represents the six classified genera.

**Table 3. Taxonomic profiling of bacterial community.**

| Phylum | Class | Order | Family | Genus | MEAN (%) ± SD | MAX (%) | MIN (%) | Prevalence (%) |
|---|---|---|---|---|---|---|---|---|
| Firmicutes | Bacilli | Bacillales | Staphylococcaceae | *Staphylococcus* | 17.0 ±23.1 | 89.3 | 0.110 | 100 |
| Actinobacteria | Actinobacteria | Actinomycetales | Corynebacteriaceae | *Corynebacterium* | 11.0 ±14.3 | 58.7 | 0.064 | 100 |
| Proteobacteria | Gammaproteobacteria | Pseudomonadales | Pseudomonadaceae | *Pseudomonas* | 3.5 ± 6.9 | 41.9 | 0.020 | 96 |
| Firmicutes | Bacilli | Bacillales | Bacillaceae | *Bacillus* | 1.89 ± 11.4 | 94.9 | 0.010 | 96 |
| Proteobacteria | Gammaproteobacteria | Pseudomonadales | Moraxellaceae | *Moraxella* | 9.1 ± 21.8 | 87.7 | 0.007 | 93 |
| Actinobacteria | Actinobacteria (class) | Actinomycetales | Propionibacteriaceae | *Propionibacterium* | 0.5 ±0.9 | 6.5 | 0.010 | 89 |
| Bacteroidetes | Flavobacteria | Flavobacteriales | Flavobacteriaceae | *Flavobacterium* | 2.36 ± 4.6 | 22.5 | 0.009 | 60 |
| Actinobacteria | Actinobacteria | Actinomycetales | Micrococcaceae | *Micrococcus* | 0.47 ± 0.67 | 3.732 | 0.008 | 52 |

Mean, Max and Min percentages as well the prevalence was estimated in the R-software on 2363 OTUs picked by MG-RAST

followed by *Moraxella* (RA-12.5%) and, *Staphylococcus* (3%). *Flavobacterium*, *Myroides*, *Pseudomonas* and *Finegoldia* although always detected, had widely varying RA. Apart from these, there were a good number of other genera (RA) ≤ 1% that are also supposed to be contributors to the overall seasonal variations (Fig 4).

According to the R analysis, the genera distribution was never the same in any of the seasons. The top 25 were plotted on bar graphs (Fig 5A–5D). It is quite evident that seven prevalent genera were repetitively detected in all four seasons. Among these, *Staphylococcus* showed maximum MD in autumn, spring and winter followed by *Corynebacterium* and *Moraxella*. Apart from this key indicator genera for autumn were *Bosea*, *Myroides Ensifer* and *Paenibacillus* with an abundance ≥ 95%. In winter, *Cytophaga*, *Agrobacterium*, and *Rhizobium* existed in 95% of

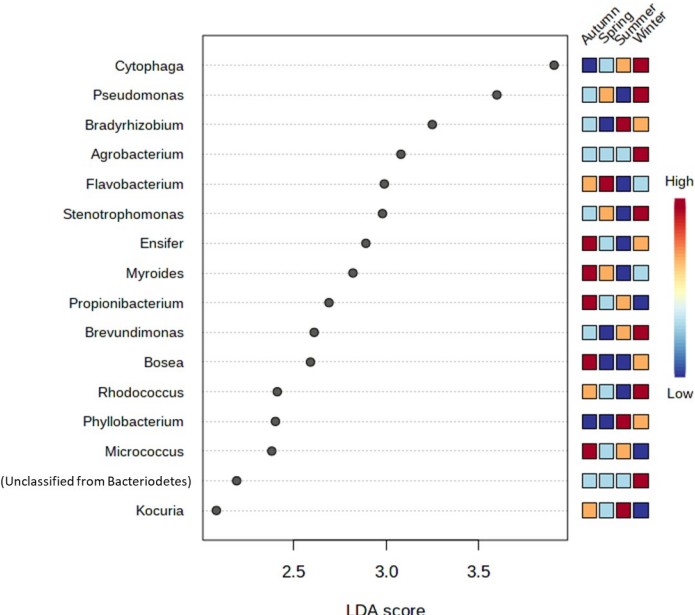

**Fig 3. Dot plot representing significantly differential genera of the nasal microbiome of health care workers of a clinical research laboratory across the four seasons.** The LEfSe algorithm applying the non-parametric factorial Kruskal-Wallis (KW) sum-rank at an FDR (q) ≤ 0.05 followed by the linear discriminant analysis (LDA) was employed to pick significantly differential taxa. Data normalization for LEfSe analysis was done through relative log expression (RLE). The heat map on the right side of the figure explains the abundance of the genus in autumn, summer, winter and spring seasons. The three colors *viz.* Blue -Beige and Red represent Low, medium, and high abundance, respectively.

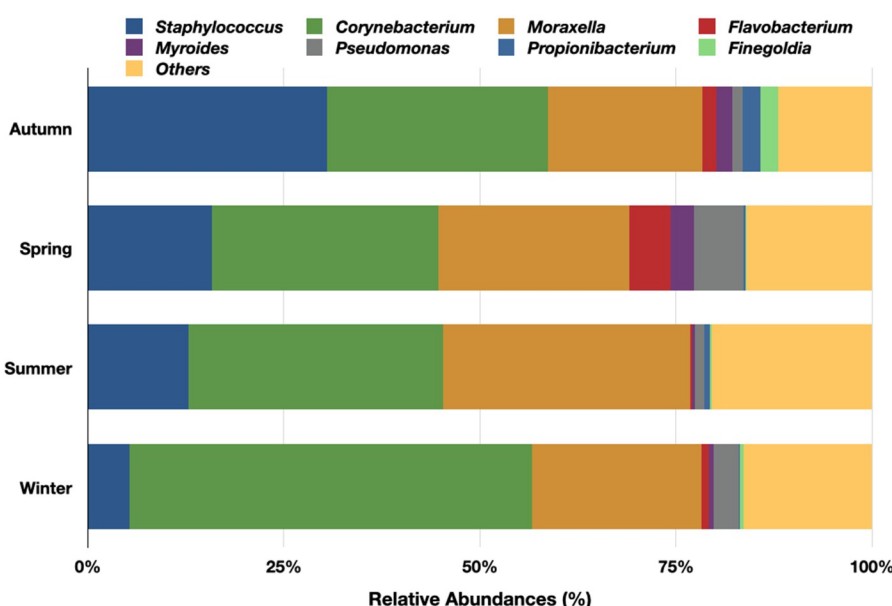

**Fig 4. Seasonal variations in the relative abundances of nasal bacterial communities of health care workers of a clinical research laboratory.** The predominant genera with RA >1.0% are presented as stacked bar plots. Genera with RA < 1.0% are combined as Others. The relative abundances are plotted on the X-axis.

samples. Spring was highly variable but none of the genera exhibited a distribution above 85%. In the summer season, *Anaerococcus* was detected in all the samples. Our results were in partial

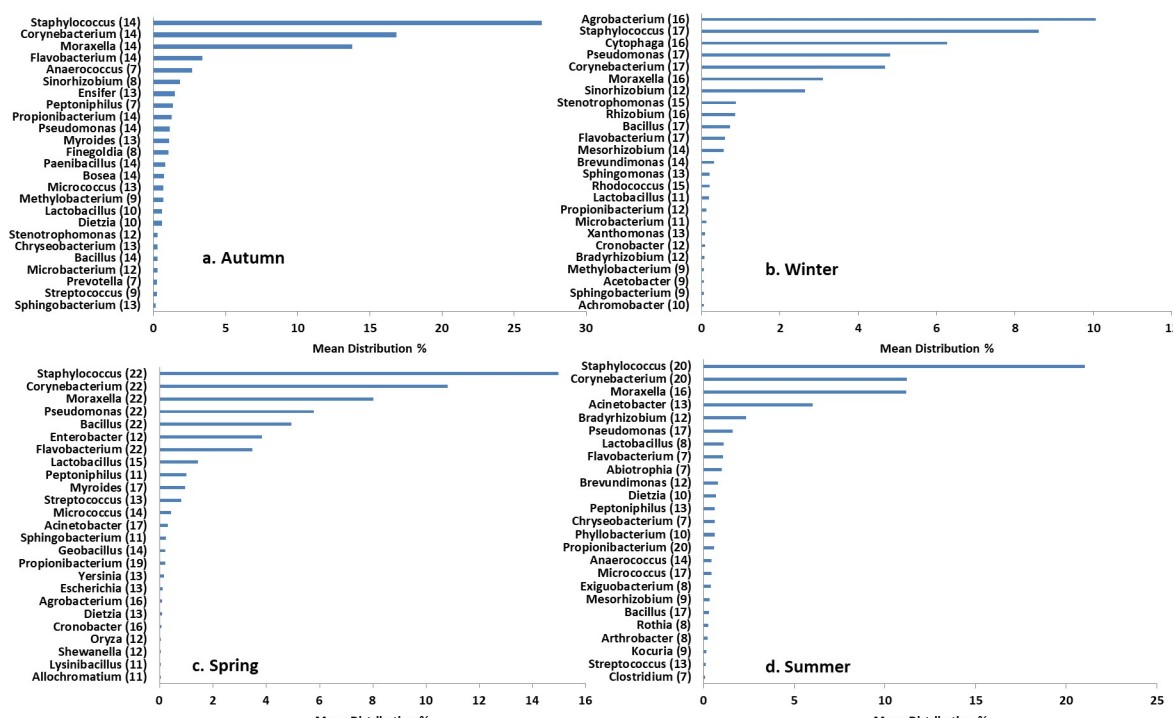

**Fig 5.** Bacterial genera of the nasal cavity of health care professionals of a clinical research laboratory in (a) Autumn; (b) Winter (c) Spring and (d) Summer. The graphs depicting the mean distribution (MD) of bacterial species were plotted in ggplot. The longer the bar, the higher the abundance. Names of the genera are plotted on the left-hand side. The values in the bracket indicate the number of samples analyzed in a particular season.

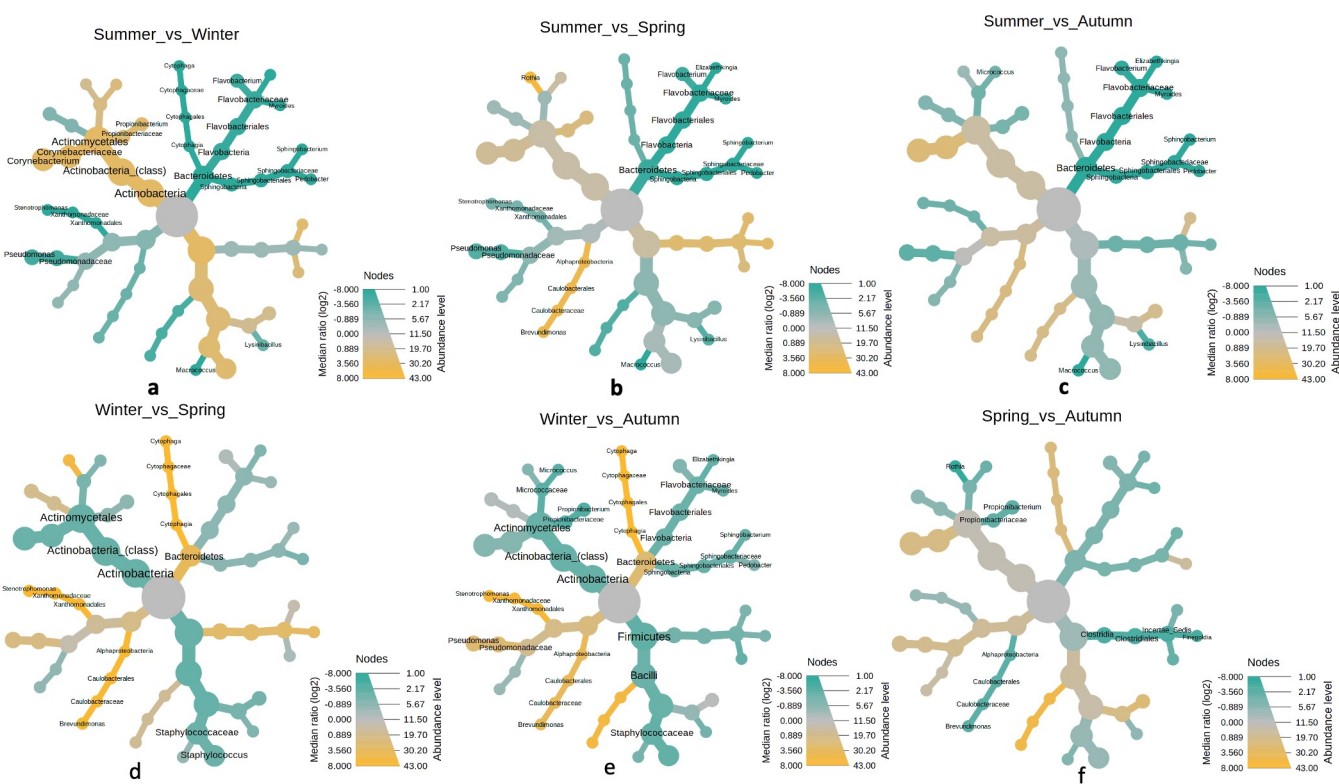

**Fig 6.** Pairwise comparison of differential abundances of bacterial taxa across the four seasons (a) summer vs winter (b) summer vs spring (c) summer vs autumn (d) winter vs spring (e) winter vs autumn and (f) spring vs autumn. Fungal communities. Only the significantly different taxonomic ranks are shown on the tree. The statistical parameter of Wilcoxon rank test (P < 0.05) was applied at median abundances to perform the differential tree analysis.

agreement with the LDA analysis. This led us to infer that the temporal variations in a surrounding favored or opposed the growth of certain forms apart from the prevalent forms. Most likely the presence of other genera alters the RA of the dominant genera.

The differential abundance analysis among the samples across the four seasons revealed significant differences. A comparison of summer versus winter revealed the genera *Cytophaga* (0.000), *Corynebacterium* (0.010), *Stenotrophomonas* (0.000), *Macrococcus* (0.002), *Lysinibacilus* (0.002), *Pedobacter* (0.005), *Sphingobacterium* (0.001), *Myroides* (0.000) and *Flavobacterium* (0.002) to be differentially abundant (Fig 6A). In summer versus spring the genera such as *Elizabethkingia* (0.003), *Myroides* (0.000), *Flavobacterium* (0.000), *Sphingobacterium* (0.000), *Pedobacter* (0.000), *Lysinibacillus* (0.000), *Macrococcus* (0.000), *Brevundimonas* (0.007), *Pseudomonas* (0.001), *Stenotrophomonas* (0.035), and *Rothia* (0.037) were significantly differentially abundant (Fig 6B). Differences in genera distribution were also observed while comparing summer versus autumn. These genera included *Elizabethkingia* (0.000), *Myroides* (0.000), *Sphingobacterium* (0.000), *Pedobacter* (0.000), *Lysinibacillus* (0.000), *Macrococcus* (0.000) and *Micrococcus* (0.040) (Fig 6C). The differential genera between winter versus spring were *Cytophaga* (0.000), *Staphylococcus* (0.031), *Brevundimonas* (0.000), and *Stenotrophomonas* (0.000) (Fig 6D). In winter versus autumn comparison genera such as *Elizabethkingia* (0.005), *Sphingobacterium* (0.002), *Brevundimonas* (0.001), *Pseudomonas* (0.018), *Stenotrophomonas* (0.000), *Micrococcus* (0.001), and *Cytophaga* (0.000) were significantly differentially abundant (Fig 6E). In spring versus autumn comparison, only three genera of *Finegoldia* (0.022), *Brevundimonas* (0.000), and *Rothia* (0.029) differentially abundant (Fig 6F).

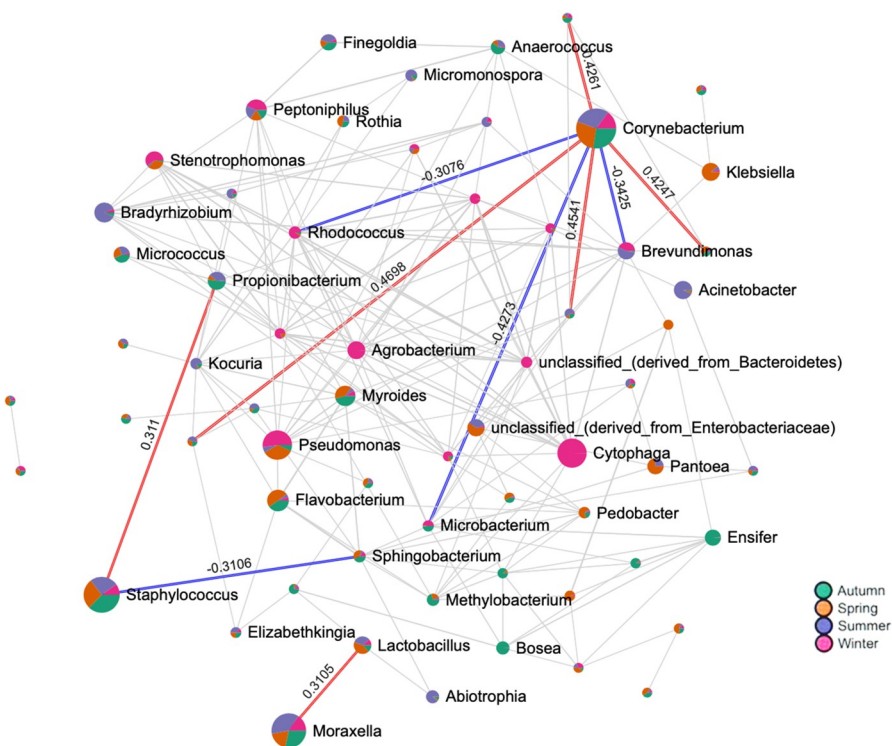

**Fig 7. A network analysis of nasal bacterial genera of health care professionals of a clinical research laboratory.**
The figure shows networks between abundant sequences at the genus level built from SparCC correlation coefficients.
Each node represents a bacterial genus and the edges represent the correlation coefficients between the genera. Blue
edges are negative correlations and red edges mean positive correlations. The values on the edges signify the
correlation coefficients. Nodes are colored according to the season (Green-Autumn; Orange-Spring; Blue-Summer;
Pink-Winter).

The SparCC network analysis returned a complex structure depicting the interactive associations of the prevalent genera with others (Fig 7). We observed that *Corynebacterium* had a positive and negative correlation with seven genera. *Staphylococcus* was positively and negatively correlated with two genera each. *Moraxella* was positively correlated with *Lactobacillus*. The colour of the nodes in the lattice indicated the occurrence of specific genera in a particular season. We extended our understanding of correlations of these three with 25 genera applying the same algorithm (Fig 8A–8C). We observed that *Corynebacterium* was positively correlated with 12 genera. It was negatively correlated with 13 of them. *Staphylococcus* was positively and negatively correlated with 10 and 15 genera, respectively. Similarly, *Moraxella* exhibited positive correlations with 14 and negative correlations with 11 genera. The positive correlation coefficients extended between 0.0–0.5 for *Corynebacterium* and *Staphylococcus* whereas for *Moraxella* it was between 0.0–0.25. Similar values for negative correlations were recorded. It was also noticed that the RA of the other genera varied (high, low, medium) in all the seasons in correlation with the RA of the prevalent taxa.

We also compared the indices of alpha diversity within bacterial populations in four different seasons and revealed significant variations ($P = 0.000$ for Observed, Chao1, ACE, and Fisher; $P = 0.010$ & $0.018$ for Shannon and Simpson, respectively). The results of the alpha diversity analysis were in agreement with our previous observations. Both the species count (Observed) and richness (Chao1, ACE, Fisher) were highest in autumn and lowest in the summer season. The species

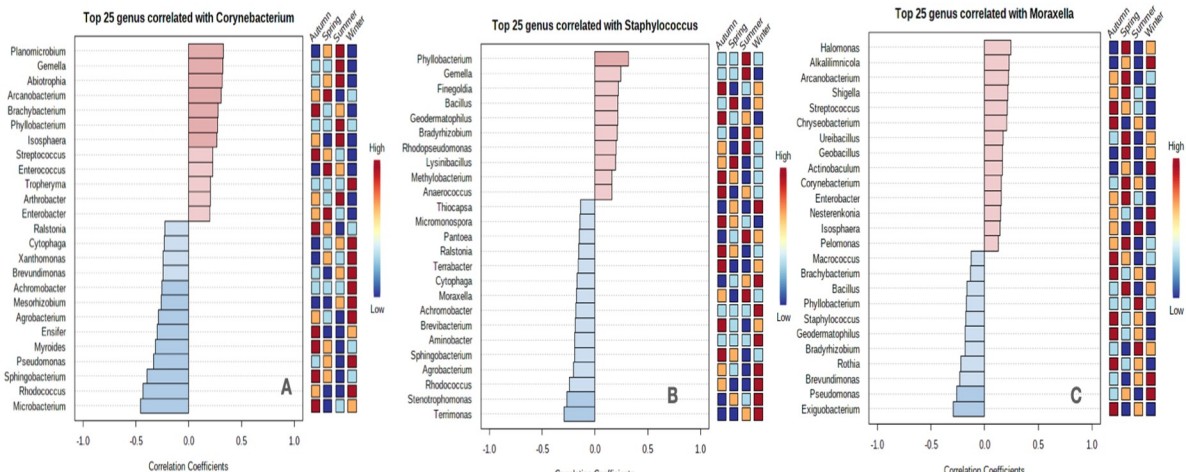

**Fig 8.** Pattern search analysis through SparCC showing positive and negative correlations of three pre-dominant genera (A) *Corynebacterium*, (B) *Staphylococcus* and (C) *Moraxella*. Correlations among the top 25 genera are presented on the bar plots. The genera are ranked by their correlation coefficients. The pink bar represents positive correlations and the blue bar depicts negative correlations. The deeper the colour the stronger the correlation. The correlation coefficients are plotted on the X-axis. A heatmap on the right shows the RA of the genera in each season (high-red; low-blue).

evenness (Shannon & Simpson) however showed a slightly different pattern that is maximum in autumn but minimum in spring. Hence, we concluded that autumn was a season that supported a more diverse growth and therefore increased counts and richness were recorded. Summer was the least preferred season for the bacterial community in terms of richness and counts. It appeared as if the less diverse population enjoyed a rich growth in summer (Fig 9). Spring and winter provided sub-optimal conditions for bacterial reproduction and multiplication.

## Intra-personal diversity

We further explored the variations in bacterial communities at the intra-personal level. For this, we employed a cohort of 10 individuals (P01-P10) who were consecutively sampled at three different time points (November-December-February). Marked differences were observed in the RA of bacterial genera (S5 Table in S1 Appendix). None of the individuals possessed the same bacterial profile in all three months (Fig 10). Especially, the RA of three prevalent genera in a particular individual differed at different time points. Further, we noticed that a specific genus was enriched in a particular individual in the respective month. For instance, P01 had the highest RA of *Staphylococcus* in November, February, and December. P02 had extremely abundant

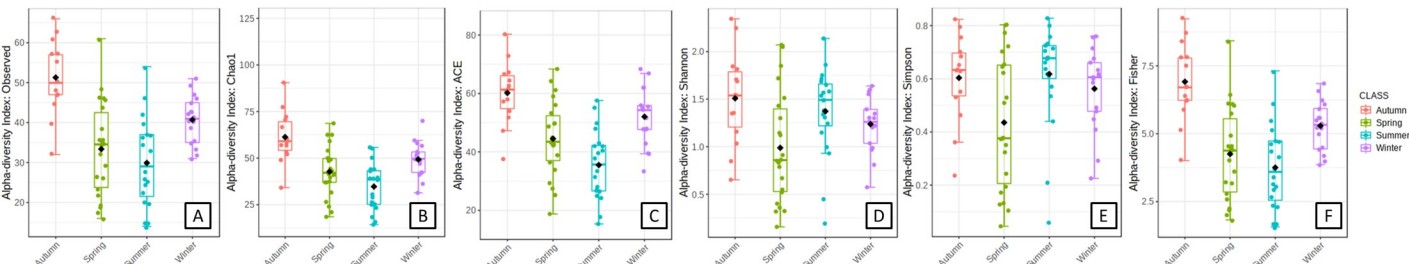

**Fig 9.** Comparison of alpha diversities of nasal bacterial communities of health care professionals stationed in a health care laboratory (A) Observed (B) Chao1 (C) ACE (D) Shannons (E) Simpsons (F) and Fisher. The boxes represent the range of alpha diversity within the samples in a particular season. Corresponding alpha diversity values are plotted on Y-axis. All the comparisons were done employing one way ANOVA at $p \leq 0.05$.

*Moraxella* in November and December that was replaced by *Pseudomonas* in February. *Staphylococcus* was dominant in P03 in November, whereas *Corynebacterium* topped the list in December and February. *Pseudomonas* was the enriched genera of P04 in all three months. The individual P05 had *Moraxella*, *Staphylococcus*, and *Peptonophilus* dominating in November, December, and February, respectively. For P06, *Pseudomonas* was the highly prevalent genus in all three months. P07 and P08 were enriched with *Moraxella* and *Staphylococcus* respectively. P09 had the most variable profile, with *Flavobacterium* prevailing in November and February. P10 again had *Pseudomonas* in abundance in November, replaced by *Bacillus* in December and *Corynebacterium* in February. Another interesting observation was the widely variable other categories possibly making a significant contribution towards the overall bacterial diversity. The distribution of all the genera within a subject is shown in Fig 11. We also performed pairwise comparison on the differences between the RA within each subject at three time points. All the subjects showed significant differences (P one tail = 0.000; P two tail = 000). The results are presented in Table 4.

From these observations, we assumed that an individuals' nasal microbiome is largely defined by the prevailing season in addition to personal habits and daily routine. We corroborated the above findings with the measurement of alpha diversity as well. All six parameters depicted wide variations in species counts, richness, and evenness during the entire sampling period (Fig 12). The difference in observed species counts varied between 1 to 2.1 folds in two consecutive months in a particular individual. In terms of species richness, the Chao1 and ACE ranged from 0.9 to 2.0 folds and 1.0 to 2.0 folds respectively. The species richness and evenness as estimated by the Shannon differed from a minimum of 0.0 to the maximum 6.8 folds. Likewise, variations were recorded for Simpson (Min. 0.0 to Max.10.5) and Fisher (Min. 0.0 to Max. 2.5). The temperature on the day of sample collection was the major environmental variable an individual experienced during these three months. Apart from this personal hygiene, home environment, routine activities at home, habits etc. are the possible sources of variation in an individual. From this, we hypothesize that each individual's microbiome is defined by the temporal variations and the day-to-day activity near his or her surroundings.

## Community structure profiling

Community structure profiling of all the samples through hierarchical clustering and dendrogram analysis (on Euclidean distances) returned profusely branched trees (S5 and S6 Figs in S1

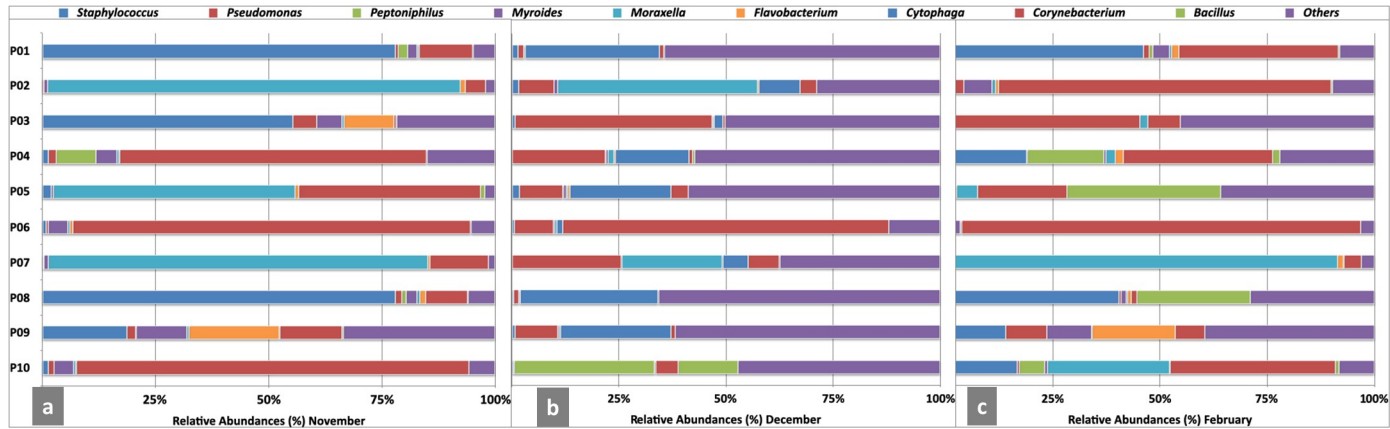

**Fig 10. Intra-personal variations in the relative abundance of bacterial genera of the nasal cavities of the health care professionals sampled consistently for three months.** P01-P10 on the Y-axis represents the ten individuals. The stacked bar shows the top ten genera with RA >1.0% in each individual and the corresponding RA are plotted on the X-axis. All the genera with RA< 1.0% are collectively presented as Others.

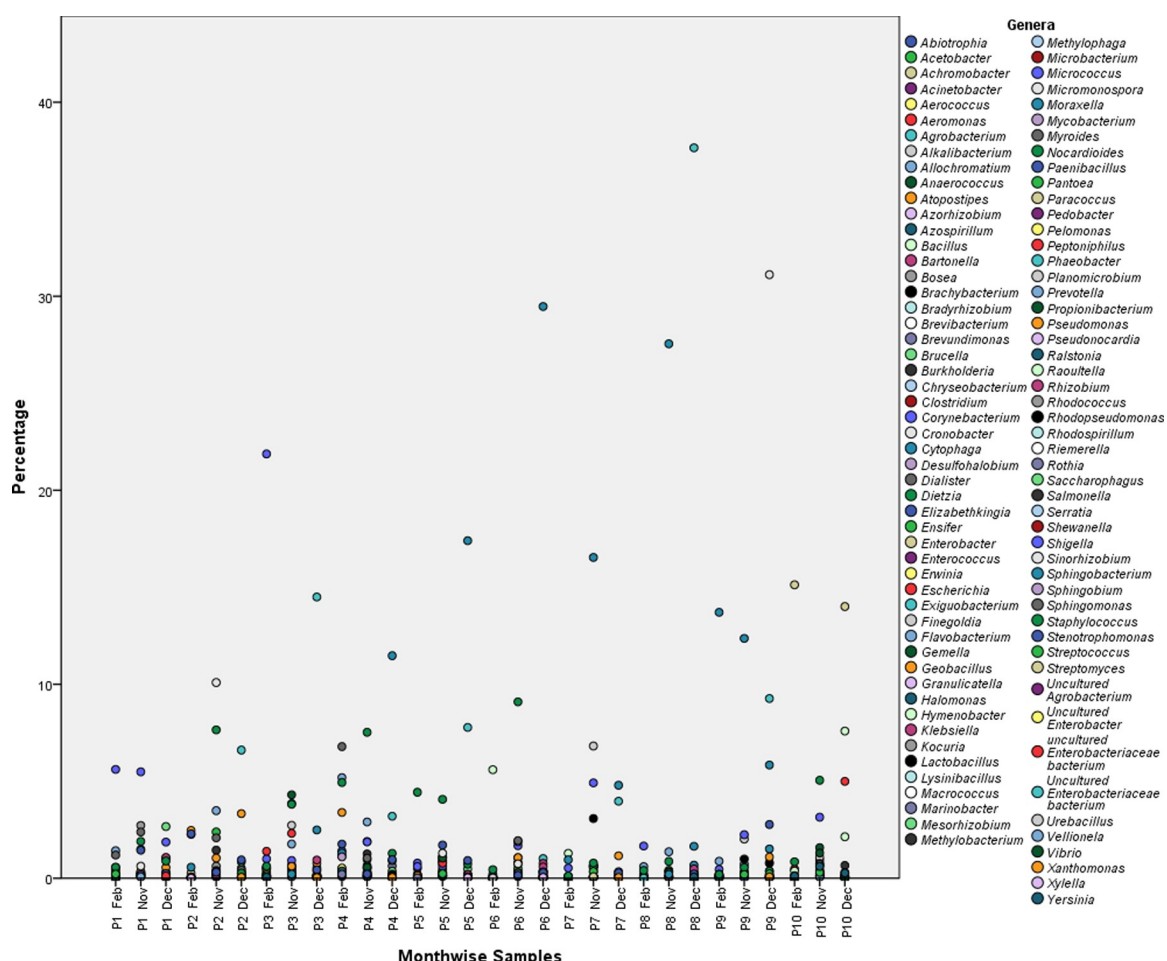

**Fig 11. Distribution of all the genera in a cohort of ten individuals at three different time points.** The percentage (RA) are plotted on the Y-axis for the corresponding subject plotted on X-axis.

Appendix) suggestive of variations between all the samples. Further to accurately identify the major experimental factor behind these variations we performed the more sophisticated beta diversity analysis. The ANOSIM coefficients ($R^2$) were evaluated on Bray Curtis distances and the results on a PCoA plot revealed four partially overlapping clusters. While choosing seasonality as an experimental factor a significantly diverse ($R^2$-0.375; $P < 0.001$) population structure was derived (Fig 13A). The variations at the 1st, 2nd, 3rd, axes, were 22.5%, 14.4% and 11.5%, respectively. We noticed that the winter and autumn clusters were distantly apart. On the contrary spring, autumn and summer were closer to each other. Spring and autumn overlapped with each other (Fig 13B). These observations suggest that most likely during extreme temperatures as in winter and summer the bacterial communities diverge, whereas the moderate temperature regimes of spring and autumn might be conducive for the specific bacterial population.

We also studied the bacterial community composition based on age, gender, and ethnicity (S7 Fig in S1 Appendix) and found neither of these parameters plays a significant role in defining the nasal bacterial community structure. Two clusters completely overlapping each other, appeared on the PCoA plot ($R^2$ 0.005; $P < 0.538$). The variations along the three axes, X, Y, and Z were 24.6%, 13.8%, and 10.1% respectively. Similar observations were recorded while

**Table 4. Pairwise comparison (Students t-test) of intra-personal diversity at three time points.**

| | P1 | | P2 | | P3 | | P4 | | P5 | | P6 | | P7 | | P8 | | P9 | | P10 | |
|---|---|---|---|---|---|---|---|---|---|---|---|---|---|---|---|---|---|---|---|---|
| | *Genera* | *Month* | *Genera* | *Month* | *Genera* | *Month* | *Genera* | *Month* | *Genera* | *Month* | *Genera* | *Month* | *Genera* | *Month* | *Genera* | *Month* | *Genera* | *Month* | *Genera* | *Month* |
| **Mean** | 52 | 0.13 | 52 | 0.18 | 52 | 0.25 | 52 | 0.24 | 52 | 0.17 | 52 | 0.20 | 52 | 0.17 | 52 | 0.25 | 52 | 0.30 | 52 | 0.22 |
| **Variance** | 887 | 0.32 | 887 | 0.82 | 887 | 2.44 | 887 | 1.06 | 887 | 1.32 | 887 | 3.21 | 887 | 1.27 | 887 | 7.04 | 887 | 4.63 | 887 | 1.76 |
| **Observations** | 309 | 309 | 309 | 309 | 309 | 309 | 309 | 309 | 309 | 309 | 309 | 309 | 309 | 309 | 309 | 309 | 309 | 309 | 309 | 309 |
| **HMD** | 0.00 | | 0.00 | | 0.00 | | 0.00 | | 0.00 | | 0 | | 0.00 | | 0.00 | | 0.00 | | 0.00 | |
| **df** | 308 | | 309 | | 310 | | 309 | | 309 | | 310 | | 309 | | 313 | | 311 | | 309 | |
| **t Stat** | 30.6 | | 30.6 | | 30.5 | | 30.5 | | 30.6 | | 30.5 | | 30.6 | | 30.4 | | 30.4 | | 30.5 | |
| **P(T< = t) one-tail** | 0.00 | | 0.00 | | 0.00 | | 0.00 | | 0.00 | | 0.00 | | 0.00 | | 0.00 | | 0.00 | | 0.00 | |
| **t Critical one-tail** | 1.65 | | 1.65 | | 1.65 | | 1.65 | | 1.65 | | 1.65 | | 1.65 | | 1.65 | | 1.65 | | 1.65 | |
| **P(T< = t) two-tail** | 0.00 | | 0.00 | | 0.00 | | 0.00 | | 0.00 | | 0.00 | | 0.00 | | 0.00 | | 0.00 | | 0.00 | |
| **t Critical two-tail** | 1.97 | | 1.97 | | 1.97 | | 1.97 | | 1.97 | | 1.97 | | 1.97 | | 1.97 | | 1.97 | | 1.97 | |

HMD-Hypothesized Mean Difference; df-degrees of freedom

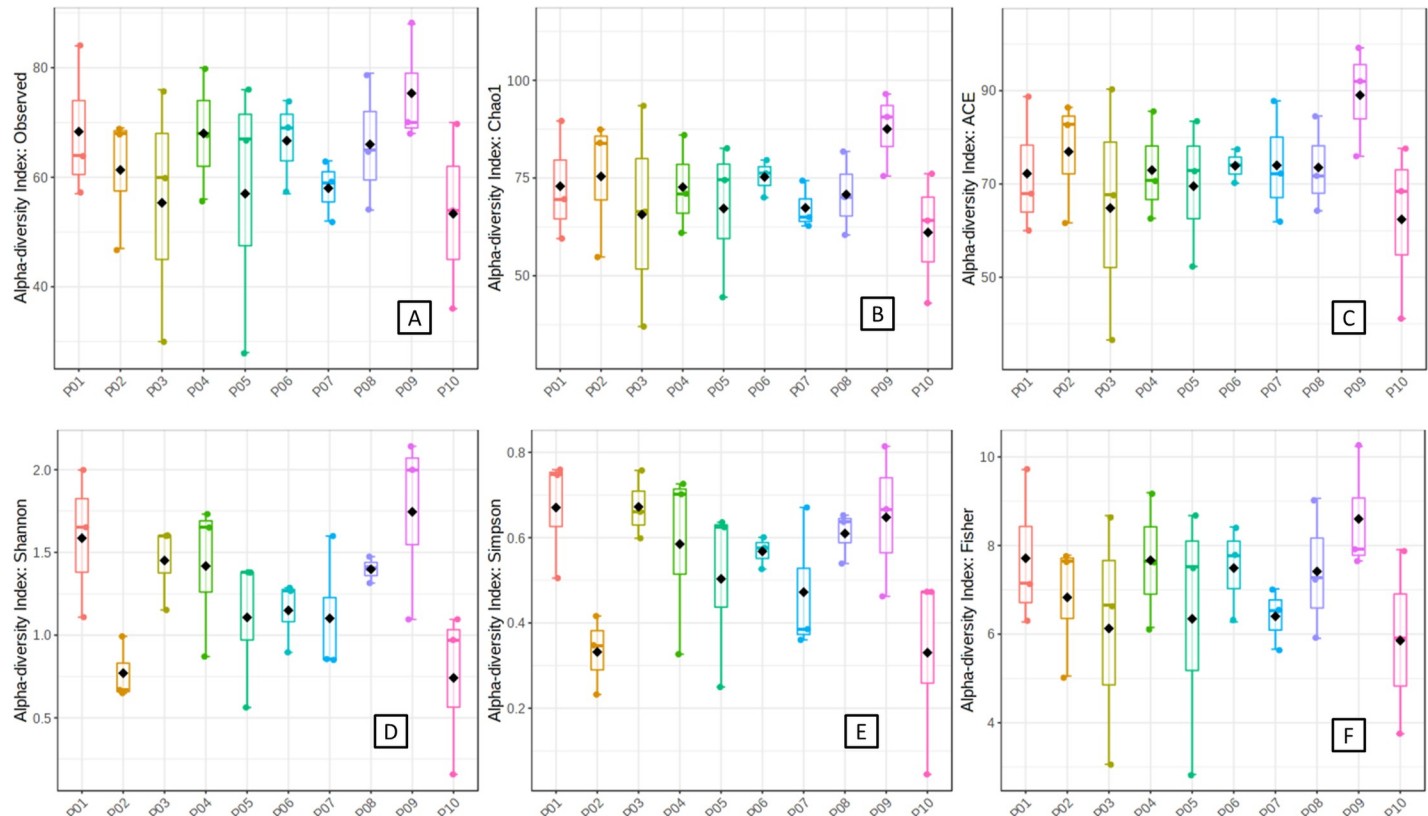

**Fig 12.** Alpha diversity indices of nasal bacterial population of ten health care professionals at three different time points (A) Observed (B) Chao1 (C) ACE (D) Shannons (E) Simpsons and (F) Fisher. The data were compared by one way ANOVA at $p \leq 0.05$. P01-P10 plotted on the X-axis represents the ten individuals sampled consecutively. Corresponding alpha diversity índices are plotted on the y-axis.

selecting gender ($R^2$ 0.019; $P < 0.216$) and ethnicity ($R^2$ 0.007; $P < 0.545$). Therefore, we concluded that in a closed laboratory setting all the demographic factors are subjugated by the prevailing working condition. The composition of the microbial community in the current study was majorly defined by seasonal variations. Other factors such as age, ethnicity, or gender

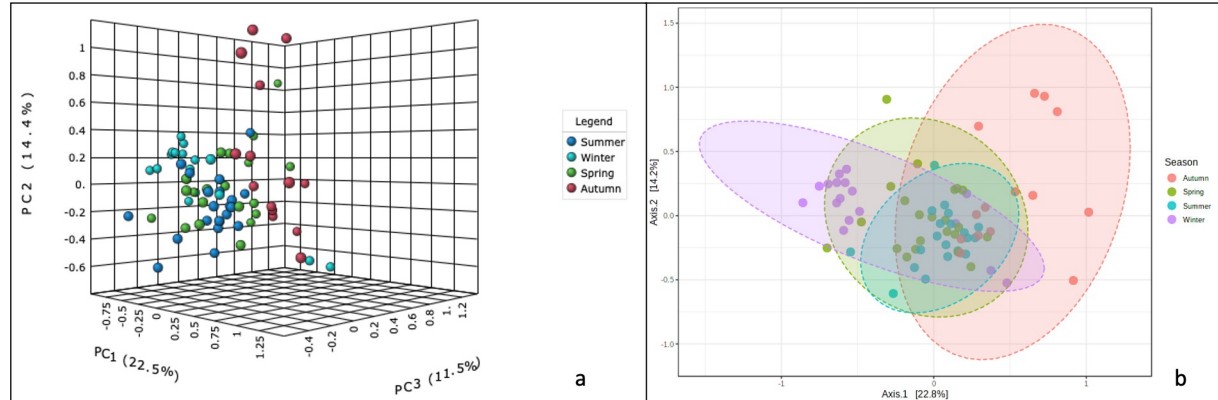

**Fig 13. Community structure profiling of nasal bacterial communities across four seasons.** (a) 3D and (b) 2D PCoA plots. Beta diversity PCoA plots were obtained by applying the ANOSIM algorithm on the Bray Curtis distances of bacterial genera. Four partially overlapping clusters representing the seasons (colors specified in the legends on the right-hand side panel).

contributed to a lower extent. Our assessment of the LDA analysis based on age, gender and ethnicity also did not return any significant features indicating their minimal role in the composition of bacterial communities of health care professionals of a clinical research laboratory in Kuwait.

## Discussion

In the present study, we report for the first time the nasal bacteriome of healthy subjects stationed in the OMICSRU/RCF, a central clinical research laboratory in the Health Sciences Center of Kuwait University. Health care professionals are often at high risk of cross-contamination. Although their physical health status seems to be perfect, they might often be carriers of pathogens. Their role in transferring these undesirable microbes to a wider population is well known [8, 46–49].

The taxonomic profiling reported almost 37% of bacteria as unclassified in the current study. Allen and co-workers reported 20% of bacteria to be unclassified [50]. Contrastingly, only 2.2% of bacteria remained unclassified below the phylum level in the nares microbiome composition of infants [51]. Owing to the geographic location novel forms may exist. This remains a subject of interest and provides a lead to identify the cryptic forms through the shotgun metagenomic analyses [52]. Further, in agreement with our findings, Proteobacteria was the dominant phyla and Gammaproteobacteria the foremost class in second-year medical students from the University of Cartagena, Colombia [49]. In partial agreement with the present investigation, the 4 most predominant phyla were reported to be Proteobacteria > Firmicutes > Bacteroidetes > Actinobacteria [53, 54]. On the contrary Actinobacteria and Firmicutes and to a lesser extent Phyla Proteobacteria and Bacteroidetes were observed in infants and their primary caregivers [51]. At subsequent levels *i.e.* class, family, and orders differing abundances were recorded by several groups as reviewed by Lee and co-workers [47]. Cross study comparisons of bacterial populations are difficult owing to variations in sampling techniques, laboratory protocols, bacterial primer selection, sequencing methods, and data analysis pipelines. Taxonomic classification (Phyla-Class-Order-Family-Genus) in our samples typically followed the classical distribution of human nose inhabiting bacteria [2, 4, 55].

Through the taxonomic profiling, core microbiome [42] and differential abundance heat tree analysis, we reported *Corynebacterium*, *Staphylococcus*, and *Moraxella* as the most stable genera in the nostrils of health care workers in a clinical laboratory. In agreement with our inferences, a study that compared the genus distribution between hospitalized and non-hospitalized health care professionals in Taiwan revealed the dominance of *Staphylococcus* in the former and *Corynebacterium* and *Moraxella* in the latter [15]. Our volunteers worked in a non-hospitalized setting and therefore, the dominance of *Corynebacterium* is well justified. Another group also showed *Staphylococcus*, *Corynebacterium*, and *Moraxella* to be present in the nasopharynx of healthy infants and identified the core microbiome as a determinant for infection spread to the lower respiratory tract [56]. *Staphylococcus* was the prevalent genus of the nostril in the reports published by Bassis and his team [57] who hypothesized that nasal cavity microbiota might be a proxy for sinusitis microbiota. The nasal cavity provides a conducive atmosphere for some unique and some commensal bacteria such as *Staphylococcus* and *Corynebacterium*, however, an imbalance in their community structure was predicted to be the driving force behind respiratory disorders such as chronic rhinosinusitis [54]. These three genera were also reported in the nasal cavities of non-healthcare workers [15, 29]. Additional investigations up to species and strain levels would provide more insights into the pathogenic status of these genera.

The exploratory investigations of the genera distribution provide a lead towards further research to be undertaken at the species level with functional annotation of pathogenic genes.

It is the bacterial species that interact with the host to establish a beneficial, commensal, or pathogenic relationship [58]. From health, perspective, further understanding of network dynamics between these taxa and other microbial communities such as fungi and viruses is recommended [12, 31]. Knowledge of the viability status of these should be taken up as metabolically active microorganisms are only potentially lethal. Studies conducted on the live bacterial component in indoor air samples in Kuwait suggested the presence of pathogenic microbes in hospitalized settings [13, 14, 59]. Inhalation of such microbes is known to be the cause of nosocomial outbreaks. The anterior nares represent the main ecological niche for *S. aureus* [49]. In the present investigation, *Staphylococcus* was found in about 75% of samples. Asymptomatic carriage of the species in health care workers is commonly documented [60–62]. Recently, a report identified 28% *S. aureus* carriage in medical students [49]. In a cohort of infants and their caregivers *S. aureus* (29–80% of subjects) was consistently present in the infants [51]. A few patterns observed in some previous studies frequently identified *P. acnes*, *S. epidermidis*, *S. aureus*, and *Corynebacterium spp.* as prevalent and abundant species in healthy controls [63–65].

The nasal ecological niche is influenced by the season and the climate of a particular region [42, 53]. Seasonality was described to be an important driving factor behind the variations in the bacterial populations in several previous reports [27, 54–56, 66, 67] as well as in the current investigation. The colonization of bacterial communities depends on the interactions with the host microenvironment which is affected by meteorological changes [58]. Among the prevalent genera, it exerted a greater effect on their relative abundances. These results are consistent with the findings of Peterson and co-workers who suggested persistence of phyla Actinobacteria, Firmicutes and, Proteobacteria [51]. However, for Others, the effect was expressed not only on the relative abundances but also on their prevalence. Perhaps the host immunity triggers a response to balance the nasal microbiome in a specific season to protect against the onset of respiratory or invasive infections. This was well demonstrated that a potential reciprocal relationship between *Corynebacterium* sps, *S. aureus*, and other players reaches an equilibrium to maintain a healthy state [68]. These correlations were observed in our study as well.

Variations within the samples were also recorded in the present study. Higher alpha diversity values in autumn are likely because of the multiplication of bacteria in the nasal cavity. Spring and summer maintained a sub-optimal count, evenness, and richness. A variable range of mean Chao1 and Shannon scores was demonstrated between the nasal communities at different time points in infants and their caregivers [51]. This diverse nature of the microbiome may be attributable to localized factors such as temperature and humidity in the respiratory tract concerning the external environment as the anterior nares are continually exposed to it [69]. A recent finding on the sinonasal bacterial microbiome suggested that species richness and diversity in bacterial populations reduce the chances of expressing a diseased condition of chronic rhinosinusitis [60]. In a country like Kuwait, autumn is the season witnessing increased hospital admissions concerning respiratory illnesses [70]. It can be thus expected that health care professionals being continually exposed to the pathogenic microbes, the natural body immune response triggers the assemblage of a richly diverse bacterial community to combat the probable incidence of infection. Richly diverse commensal microbiota is crucial for sustaining an equilibrium in the bacterial community, maintaining the integrity of the mucosal barrier, and many other aspects of health, such as resistance to infection, an effective immune system and, favorable nutritional status [15]. The air quality of the external atmosphere considerably affects the microbiological component of the indoor environment. Kuwait is a country with high dust loading. Several pathogenic and non-pathogenic microbes have been previously demonstrated to be associated with different size fractions of dust in the country [11, 31]. The incidences of allergy and asthma are highly prevalent in this country. A study

conducted on aeroallergens and its relationship with asthma-related visits to hospitals in Kuwait [70, 71] established increased admission due to elevated pollen counts in the air.

Intra-individual variations of the nasal microbial communities of health care professionals have been studied to a very limited extent. There is a pressing need for longitudinal studies to examine the stability of nasal microbiota, as any shifts in the stable communities' leads towards diseased states [48]. Owing to the confinement of the study subjects in a closed indoor laboratory setting and conclusions from previous studies [55, 72] we hypothesized limited variability to be present within the participants in the present study. On the contrary, our findings suggest wide variability intra-individually in terms of changed time points. Our results were consistent with some other reports depicting bacterial densities to be altered in the same individuals in fall and summer [53], mono and dizygotic twins [67] and post-operational treatment cessation [73]. Also at the intra-personal level, some communities persisted and others fluctuated [48]. This can be attributed to diet, environment, host genetics, daily microbial exposure and personal habits [55, 58, 74, 75]. We also credit the changes in the RA of the constantly detected genera to be environmentally influenced. These genera most likely experience a natural shift in diversity with time and space [48] as post working hours each individual is exposed to the external atmosphere. A baseline study conducted on the taxonomic profiling of the airborne microbial communities of the external atmosphere demonstrated the occurrence of a variable microbial population in the inhalable and respirable air fractions in Kuwait. These variations were due to space and time [11, 31]. This leads us to conclude that differences in RA in the same subjects between different conditions are obvious and each individuals' microflora is jointly influenced by the working atmosphere as well as the external surroundings. The indoor microbiota of pig farms was known to drive the composition of the pig farmers' nasal microbiota in a season-dependent manner [76].

Bacterial community structure is predicted to be influenced by the living environment, age, ethnicity, and gender of the cohort. In a longitudinal study, however, the roles of each of these factors are altered. Our estimates of beta diversity were in agreement with other investigations where bacterial profiles were grouped by season [51, 77, 78]. Unlike some previous studies, in the present investigation, minimal grouping was observed based on ethnicity, age, and gender [9, 55, 77]. Most of these studies have compared the groups living in separate geographical conditions. The study subjects in the present research were from the country of Kuwait with the smallest land area. The staff working in the clinical research laboratory mostly resided with a range of 5–20 km and therefore, were exposed to a more or less homogenous external atmosphere at a single time point.

## Conclusions

Health care workers in a clinical research laboratory possess a bacterial community composition of commensal and opportunistic pathogens. These genera interact with a variety of other bacterial genera to establish healthy baselines. The core bacterial genera of the nasal microbiome of health care workers vary according to the season. None of the individuals possesses a stable personalized bacteriome as proven by differences in relative abundances at different time points. The cause of the variation is largely attributed to the environmental changes in the external atmosphere. Our findings suggest the need for continuous monitoring of nasal microbiota at the health care centre's to prevent nosocomial infections.

## Supporting information

**S1 Appendix. Supplementary data for the manuscript entitled "Composition of nasal bacterial community and its seasonal variation in health care workers stationed in a clinical**

**research laboratory".**
(PDF)

## Acknowledgments

We thank Mr. Faraz Shaheed Usmani for his technical assistance.

## Author Contributions

**Conceptualization:** Nazima Habibi, Abu Salim Mustafa.

**Data curation:** Nazima Habibi, Mohd Wasif Khan.

**Formal analysis:** Nazima Habibi.

**Funding acquisition:** Abu Salim Mustafa.

**Methodology:** Nazima Habibi.

**Project administration:** Abu Salim Mustafa.

**Resources:** Nazima Habibi, Abu Salim Mustafa.

**Software:** Nazima Habibi, Abu Salim Mustafa, Mohd Wasif Khan.

**Supervision:** Abu Salim Mustafa.

**Validation:** Nazima Habibi, Mohd Wasif Khan.

**Visualization:** Nazima Habibi, Mohd Wasif Khan.

**Writing – original draft:** Nazima Habibi.

**Writing – review & editing:** Nazima Habibi, Abu Salim Mustafa.

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
