## [Decision Letter · Decision Letter 0]

24 Jun 2021

PONE-D-21-17891

Composition of Nasal Bacterial Community and its Temporal Variation in Health Care Workers Stationed in a Clinical Research Laboratory

PLOS ONE

Dear Dr. Habibi,

Thank you for submitting your manuscript to PLOS ONE. After careful consideration, we feel that it has merit but does not fully meet PLOS ONE’s publication criteria as it currently stands. Therefore, we invite you to submit a revised version of the manuscript that addresses the points raised during the review process.

Substantial revision of the English is required, with special attention to making sure terms are used accurately throughout the text. Both reviewers also raised concerns about the treatment of the control samples, which requires extensive clarification.

We look forward to receiving your revised manuscript.

Kind regards,

Christopher Staley, Ph.D.

Academic Editor

PLOS ONE

Journal Requirements:

Reviewers' comments:

Reviewer's Responses to Questions

**Comments to the Author**

1. Is the manuscript technically sound, and do the data support the conclusions?

Reviewer #1: Yes

Reviewer #2: Partly

2. Has the statistical analysis been performed appropriately and rigorously? 

Reviewer #1: Yes

Reviewer #2: I Don't Know

3. Have the authors made all data underlying the findings in their manuscript fully available?

Reviewer #1: Yes

Reviewer #2: Yes

4. Is the manuscript presented in an intelligible fashion and written in standard English?

Reviewer #1: Yes

Reviewer #2: No

5. Review Comments to the Author

Reviewer #1: 1- The sentence in page 4 line 73 was a repeat of sentence in page 3 line 46

2- The sentence in page 4 line 87-88 (below 30 years and above 30 years of age), the age range need to verified with range.

3- The design of the experiment in this work focused on the effect of indoor working environmental on nasal bacterial diversity, however, the authors does not explain or take in their account the effect both indoor environment at home and outdoor environment in their study. This can be achieved by using a control group from the same city outside the working environment, instead the authors use control group from different environment (20 healthy individuals (HC) working in a livestock farm in Iowa)

4- The authors enrolled 47 staff of the OMICS Research Unit/ Research Core Facility, Health Sciences Center, Kuwait University, Kuwait, or the staff from the departments in the same center. The indoor environment is different between different department, the author needs to explain why use individual diverse indoor environment as homogenous sample.

5- The authors need to explain why enrolled 10 individuals for intra personal diversity for three seasons (winter, spring and autumn) and does not include summer.

6- The author needs to explain the working indoor environment for the 10-individual enrolled in intra personal diversity, as none of the individual posses the same bacterial profile.

Reviewer #2: In the manuscript by Habibi et. al, the authors investigate the nasal microbiome of a cohort of workers in a clinical research laboratory. They collect samples across a single year, representing a snapshot of the microbiome at that time. They also longitudinally follow a small cohort of individuals and assess intra-person variability within those people.

It is clear that the nasal niche represents an important reservoir for bacterial colonization. Moreover, the bacteria that are found there can include opportunistic pathogens that can infect those that carry them, or be spread to other individuals. Thus, the topic is interesting and important. This being said, I found the presented details to be insufficient and the data analysis and presentation to be difficult to follow. I also think the authors need to be very careful about not over interpreting their results. Specific examples are provided below.

Major comments:

1. The description of who was sampled and when is very confusing and needs to be clarified. The author’s clearly state that 47 individuals were included in the study. Of these, 10 individuals were sequentially sampled for 3 time points. Were the other 37 individuals only sampled once in one of the indicated seasons? If yes, that would suggest that the authors are looking at a total of 37+30=67 samples. However, this doesn’t match what’s presented in the paper or Table 1. Table 1 says that a total of 73 “individuals” were sampled in the 4 seasons. However, the other variables (age, gender, etc.) total to 75 “samples.” None of this makes sense and the authors need to clearly and precisely explain how many individuals were sampled, how many samples they got from each individual, etc. They also should make sure to not use the terms ‘individual’ and ‘sample’ interchangeably; these are different.

2. In the methods, the authors mention several important controls that were included in the DNA isolation, 16S amplification and sequencing to account for any contamination. However, the results of what was obtained from those controls is not mentioned anywhere in the manuscript.

3. The authors need to include a detailed section on the sequencing results. How many reads were obtained? How many were left after filtering? How many reads were there per sample (range and median). Were all of the remaining reads used or was the data rarified in some way? None of these details are currently provided.

4. The authors use the word “core microbiome” without ever really defining what that means. Are these found in 100% of samples? I don’t think so based on figure 2, which I believe suggests that some of their ‘core’ is in as little at 20% of samples? Why would this be considered ‘core’?

5. I’m not sure the authors consistently use terms like ‘relative abundance’, ‘prevalence’, etc. Please make sure to clearly define what these terms mean within your study and then be consistent with their usage. Otherwise, it’s difficult to follow the results, descriptions, and figures.

6. For the seasonal variation, were all available samples included in this analysis? In other words, did this analysis include the 30 samples that came from the 10 individuals that were sampled in the 3 sequential months? What’s included should be clearly stated.

7. The sampled individuals were listed as healthy, but were they healthy for the entire year of the study? Did any of them take antibiotics at any point within the year? If yes, this could certainly account for variability. Were there any exclusion criteria that were used to select participants?

8. The authors repeatedly state that their data show that temperature/season are the driving forces resulting in nasal microbiome changes. These statements need to be softened. Realistically, these individuals spend the majority of their time outside of the work environment and may be exposed to any number of external exposures or environments that may affect the microbiome. Thus, the authors should not overstate their results. Temperture/season seems to be important in the very small number of variables that you assessed.

9. I think that all of the legends should be expanded to very clearly explain what is shown/found in the figures and tables. Make sure your terminology is consistent and accurate.

Specific comments:

1. Line 22: Why are these individuals ‘potentially healthy’? This is an odd phrase.

2. Line 27: Were these different ‘in all 4 seasons’ or actually ‘different across the 4 seasons’. For the first to be true, you would have to observe significant differences in each season when compared to each of the other seasons. I’m not sure that’s what the data show.

3. Line 44: change to read “…tract and are involved in the….”

4. Lines 73-79 are virtually identical to what is presented in the preceding paragraphs and should be deleted or edited.

5. Line 88: I found the statement of age to be very confusing. Upon looking at the table, I realized the authors were saying that they grouped the individuals into two categories (<30 years or >30 years). They really should provide the age range of the sampled individuals and then state that they broke them into groups to assess the potential importance of age on the nasal microbiome.

6. Lines 99-101 and elsewhere: The authors should not call these ‘controls’. These are not controls, but are examples of different data sets that you compared your data to as a way to determine how comparable the results were.

7. Line 111: change to read “…processing for subsequent 16S amplification.”

8. Line 114: the ‘S’ should be capitalized in 16S.

9. Line 116-117: Do these primers have names? Please provide names in addition to sequences.

10. Line 155: the word ‘filtered’ implies that the 2363 OTUs were discarded from the analysis. I don’t think that is what the authors mean. I think they are saying that that many OTUs remained after all of the filtering steps.

11. Line 181: Do the authors really classify at the ‘species’ level? This is difficult to do with such short reads. If they were able to classify some sequences (OTUs) to the species level, I saw no real discussion of various species that they identified.

12. Lines 297-298 as compared to Lines 314-315. Aren’t these two results/statements conflicting?

13. Lines 360, 397, and elsewhere: terms like ‘disease condition’ are vague. State the disease that you are referring to.

14. Line 443: Change to read “…possesses a stable personalized microbiome….”

15. Line 460: ‘tree by’.

16. Figure 11 panels B, C and D could be moved to the supplement since panel A is the only one that showed an interesting result.

17. I think the authors should be careful with the use of the terms ‘longitudinal and temporal.’ I would agree that the 10 individuals that were sampled in sequential months were followed longitudinally. ‘Temporal’ can also imply the same thing as longitudinal, meaning you followed something across time. Maybe “seasonal” would be a better word since you don’t appear to sample most of the individuals more than once? This could be altered in the title.

6. PLOS authors have the option to publish the peer review history of their article (what does this mean?). If published, this will include your full peer review and any attached files.

Reviewer #1: **Yes: **Amro Hanora

Reviewer #2: No

---

## [Author Response · Author response to Decision Letter 0]

19 Jul 2021

PONE-D-21-17891

Composition of Nasal Bacterial Community and its Temporal Variation in Health Care Workers Stationed in a Clinical Research Laboratory

PLOS ONE

Response to Academic Editors’ Comments

Substantial revision of the English is required, with special attention to making sure terms are used accurately throughout the text. Both reviewers also raised concerns about the treatment of the control samples, which requires extensive clarification.

Thank you, Dr. Christopher Stanley, for your positive feedback and considering our manuscript for peer review. 

Your suggestions on improvement of English language have helped us significantly to improve the legibility of the text. The English language has been edited through the Grammarly tool. It has also been read by a Native English Speaker in University of Manitoba. 

We have keenly looked at the usage of terms such as relative abundance, prevalence and mean distribution and have made necessary changes wherever required for clarity of explanation. We have also provided a short definition of each term in the text.

While the experiments were conducted on health care professionals only and many of the analysis was completed after the project completion date, lack of a control group (non-healthcare professionals) was felt, we, therefore, used two publicly available datasets to compare our results with and use them as control. 

All the changes in the manuscript have been marked in red.

We hope the manuscript is now fit for publication in PLoS ONE

Response to Reviewers Comments

Reviewer #1 Prof. Amro Hanora

We are thankful to Prof. Amro, for reviewing our manuscript and providing his valuable comments. This has helped us a lot to amend our manuscript. He has raised valid concerns and we have provided our response for each of his comments as given below.

1- The sentence in page 4 line 73 was a repeat of sentence in page 3 line 46

Response: Thanks for bringing this to our notice. We have replaced this section with new text (page 4; lines 77-81).

2- The sentence in page 4 line 87-88 (below 30 years and above 30 years of age), the age range need to verified with range.

Response: We agree with the reviewer’s suggestion and have now provided the range of age groups (page 4; line 90).

3- The design of the experiment in this work focused on the effect of indoor working environmental on nasal bacterial diversity, however, the authors does not explain or take in their account the effect both indoor environment at home and outdoor environment in their study. This can be achieved by using a control group from the same city outside the working environment, instead the authors use control group from different environment (20 healthy individuals (HC) working in a livestock farm in Iowa)

Response: The reviewer has raised a valid concern. Initially during the study only health care professionals were sampled. A control group was largely lacking. However, by the time the experiments on metagenomic analysis performed and complete analysis achieved the project was over. The study seemed to be incomplete without a control group and we felt that the results obtained cannot be referred to the microbiota of health care professionals. We therefore applied a strategy to compare our data with the data deposited in online repositories on nasal microbiota. We have compared our data with two studies, one is a group of health care professionals working in health care institutes of Taiwan (Chen et al., 2019) and the second is healthy individuals working in a livestock farm (OTUS from individuals not associated with animals) in Iowa (Kate et al., 2019). We downloaded the OTU files from these respective studies and merged it with our OTU table. We applied the centred log ratio normalisation on the entire dataset and performed the taxonomic profiling, linear discriminate analysis (LefSe) as well as the beta diversity clustering (PCoA). We discovered 1 genus to be significant differentially abundant among these studies. The beta diversity analysis plotted them as three separate cluster on the PCoA plot.

4- The authors enrolled 47 staff of the OMICS Research Unit/ Research Core Facility, Health Sciences Center, Kuwait University, Kuwait, or the staff from the departments in the same center. The indoor environment is different between different department, the author needs to explain why use individual diverse indoor environment as homogenous sample.

Response: The staff belonged to different department but during the study period they were using the research core facility to conduct their experiments/use the instruments and equipment available in the facility. The OMICSRU also provides working space to its users. The volunteers sampled were working in the same indoor environment as the staff of the facility during the present investigation and therefore the indoor working environment was considered homogenous. Text describing the OMICSRU location and usage has been added to the main manuscript (page 4; lines 93-96). 

5- The authors need to explain why enrolled 10 individuals for intra personal diversity for three seasons (winter, spring and autumn) and does not include summer.

Response: Our aim was not to study the effect of seasonality among these individuals rather to see the differences in their microbiota at different time points. Three samples from the same individual at three different time points were considered as technically reasonable to study the intra-personal diversity and therefore, sampling was not done further.

6- The author needs to explain the working indoor environment for the 10-individual enrolled in intra personal diversity, as none of the individual posses the same bacterial profile.

Response: The 10 individuals worked on the first and second floor of the OMICSRU. Well, it was quite perplexing for us as well, that none of the individuals shared a common microbiome (page 18; line 440-443). We attributed this to diet, environment, host genetics, daily microbial exposure and personal habits (page 18; line 446-447). We have also added a table (Table 2) explaining the details of working locations of these individuals. 

Reviewer #2

In the manuscript by Habibi et. al, the authors investigate the nasal microbiome of a cohort of workers in a clinical research laboratory. They collect samples across a single year, representing a snapshot of the microbiome at that time. They also longitudinally follow a small cohort of individuals and assess intra-person variability within those people.

It is clear that the nasal niche represents an important reservoir for bacterial colonization. Moreover, the bacteria that are found there can include opportunistic pathogens that can infect those that carry them, or be spread to other individuals. Thus, the topic is interesting and important. This being said, I found the presented details to be insufficient and the data analysis and presentation to be difficult to follow. I also think the authors need to be very careful about not over interpreting their results. Specific examples are provided below.

We express our gratitude to the reviewer for his positive feedback and finding the topic worthy of investigation and interesting. His thorough review and critical comments has significantly improved our manuscript. We have provided a point-by-point response to all his concerns in the text below:

Major comments:

1. The description of who was sampled and when is very confusing and needs to be clarified. The author’s clearly state that 47 individuals were included in the study. Of these, 10 individuals were sequentially sampled for 3 time points. Were the other 37 individuals only sampled once in one of the indicated seasons? If yes, that would suggest that the authors are looking at a total of 37+30=67 samples. However, this doesn’t match what’s presented in the paper or Table 1. Table 1 says that a total of 73 “individuals” were sampled in the 4 seasons. However, the other variables (age, gender, etc.) total to 75 “samples.” None of this makes sense and the authors need to clearly and precisely explain how many individuals were sampled, how many samples they got from each individual, etc. They also should make sure to not use the terms ‘individual’ and ‘sample’ interchangeably; these are different.

Response: We are apologetic of our ambiguous expression on the sampling details. We have modified the Table 1 completely for more clarity. We would like confirm that the total number of samples analyzed were 73. These 73 samples were collected from 47 individuals. The 10 individuals sampled thrice are a part of the 73 samples. The breakdown of samples collected in each season is provided in the modified Table 1. An additional table (Table 2) explaining their working location has been added to the manuscript.

2. In the methods, the authors mention several important controls that were included in the DNA isolation, 16S amplification and sequencing to account for any contamination. However, the results of what was obtained from those controls is not mentioned anywhere in the manuscript.

Response: The reviewer has raised a valid concern. Concerning the DNA isolation, the readings for DNA concentration from the plain swab as estimated by the Qubit HS dsDNA assay was too low, indicating absence of DNA. This statement has been provided in the manuscript (page-5 line 121). Regarding the 16 s amplification, the bioanalyzer trails of the positive control, negative control (nuclease free water), DNA Marker and sample library have been provided in the supplementary file Fig S1. Also, the Phred score of the libraries sequenced and analyzed through FASTQC have been mentioned in the supplementary table S1. 

3. The authors need to include a detailed section on the sequencing results. How many reads were obtained? How many were left after filtering? How many reads were there per sample (range and median). Were all of the remaining reads used or was the data rarified in some way? None of these details are currently provided.

Response: We would like to bring to the reviewer’s attention that the details on sequencing reads were provided in the supplementary file Table 1. Data rarefaction was also performed and the rarefaction curves with Goods coverage was also provided in the supplementary data sheet Fig S1 (Main manuscript page 7; line168). The quality parameters of each sample can be assessed at the online MG-RAST repository. However, to give a quick glimpse on the quality parameters of the data to the readers we have added the total read counts and sequence counts to the main text in the manuscript. Page no. 8; lines 185-192.

4. The authors use the word “core microbiome” without ever really defining what that means. Are these found in 100% of samples? I don’t think so based on figure 2, which I believe suggests that some of their ‘core’ is in as little at 20% of samples? Why would this be considered ‘core’?

Response: We agree to the reviewer’s concern and we too found it difficult to set a cut-off sample size, as there was no rationale available for setting up the sample size. We followed the default settings of the software and obtained a heatmap with 12 genera showing prevalence above 20%. Very recently, the commentary published by Risely et al 2020 (new citation in the manuscript) has mentioned the sample size cut-off for host core microbiome as 30%. We have therefore, reworked this figure and modified the text accordingly (page 9; lines 211-219). The core microbiome analysis is based on genera recorded in more than 30% of the samples. The figure legend has also been modified accordingly. 

5. I’m not sure the authors consistently use terms like ‘relative abundance’, ‘prevalence’, etc. Please make sure to clearly define what these terms mean within your study and then be consistent with their usage. Otherwise, it’s difficult to follow the results, descriptions, and figures.

Response: These are the terms obtained by default through the respective software. The relative abundance refers to the zero inflated microbial composition (page 8; line 199) collectively in all the samples, the prevalence means its occurrence in number of samples (page 9; line 215). The mean distribution is more or less similar to relative abundance. We have provided an explanation of these terms in the text. 

6. For the seasonal variation, were all available samples included in this analysis? In other words, did this analysis include the 30 samples that came from the 10 individuals that were sampled in the 3 sequential months? What’s included should be clearly stated.

Response: The seasonal variation was studied for all the samples (including the 30 samples for intra-personal diversity) sampled in the 3 sequential months. The table describing the sampling details has been modified and specifies the exact number of samples for each experimental parameter.

7. The sampled individuals were listed as healthy, but were they healthy for the entire year of the study? Did any of them take antibiotics at any point within the year? If yes, this could certainly account for variability. Were there any exclusion criteria that were used to select participants?

Response: The reviewer has raised a valid concern. The word healthy here signifies that the sampled individuals did not suffer from any upper respiratory tract infections and were not on any antibiotics pertaining to it during the entire study period. In fact, higher number of volunteers (than mentioned in the manuscript) were registered during the study period and if anyone reported the intake of antibiotics, were not sampled further and excluded from the study. A statement relevant to the intake of antibiotics has been added to the manuscript text. Page no 5 lines 105-106. 

8. The authors repeatedly state that their data show that temperature/season are the driving forces resulting in nasal microbiome changes. These statements need to be softened. Realistically, these individuals spend the majority of their time outside of the work environment and may be exposed to any number of external exposures or environments that may affect the microbiome. Thus, the authors should not overstate their results. Temperture/season seems to be important in the very small number of variables that you assessed.

Response: We totally agree with the reviewer’s suggestion. We had no intentions to overstate the results. We have revisited the result and discussions section and rephrased the text to remove the impression of overstatement.

9. I think that all of the legends should be expanded to very clearly explain what is shown/found in the figures and tables. Make sure your terminology is consistent and accurate.

Response: Modification have been done as suggested

Specific comments:

1. Line 22: Why are these individuals ‘potentially healthy’? This is an odd phrase.

Response: We have removed the word “potentially” (page 2; line 25)

2. Line 27: Were these different ‘in all 4 seasons’ or actually ‘different across the 4 seasons’. For the first to be true, you would have to observe significant differences in each season when compared to each of the other seasons. I’m not sure that’s what the data show.

Response: We mean to say different across the 4 seasons. We have rephrased this sentence on page 2, line 30

3. Line 44: change to read “…tract and are involved in the….”

Response: Changed as suggested (page 3; line 48)

4. Lines 73-79 are virtually identical to what is presented in the preceding paragraphs and should be deleted or edited.

Response: We somehow overlooked the redundancy here. We have replaced the text with new information (page 3; lines 77-81). 

5. Line 88: I found the statement of age to be very confusing. Upon looking at the table, I realized the authors were saying that they grouped the individuals into two categories (<30 years or >30 years). They really should provide the age range of the sampled individuals and then state that they broke them into groups to assess the potential importance of age on the nasal microbiome.

Response: We agree to the reviewer’s response. We have rephrased our statement and specified the range of age groups categorized in two groups A and B (page 4; line 90).

6. Lines 99-101 and elsewhere: The authors should not call these ‘controls’. These are not controls, but are examples of different data sets that you compared your data to as a way to determine how comparable the results were.

Response: We have rephrased this statement and removed the word control from the text (Page 5; line 108). 

7. Line 111: change to read “…processing for subsequent 16S amplification.”

Response: Changed as suggested (page 5; line 119)

8. Line 114: the ‘S’ should be capitalized in 16S.

Response: Changed as suggested (page 6; line 123)

9. Line 116-117: Do these primers have names? Please provide names in addition to sequences.

Response: These primers were designed as per the 16S amplification application note of Illumina. However, the primers recommended by Illumina are based on the most promising primer pair S-D-Bact-0341-b-S-17/S-D-Bact-0785-a-A-21 identified by Klindworth et al., 2013. This information has been added to the text (page 6; lines 124-129). 

10. Line 155: the word ‘filtered’ implies that the 2363 OTUs were discarded from the analysis. I don’t think that is what the authors mean. I think they are saying that that many OTUs remained after all of the filtering steps.

Response: Thanks for bringing this to our notice. We have replaced the word ‘filtered’ by ‘picked'. (Page 7; line 166)

11. Line 181: Do the authors really classify at the ‘species’ level? This is difficult to do with such short reads. If they were able to classify some sequences (OTUs) to the species level, I saw no real discussion of various species that they identified.

Response: We did the profiling upto the species level, however as the 16Samplicon is not an appropriate technique to classify the sequences upto the taxon therefore, we did not mention any of these results. To avoid any confusion, we are removing this statement from the text (page 8; line 200).

12. Lines 297-298 as compared to Lines 314-315. Aren’t these two results/statements conflicting?

Response: We had performed the hierarchical clustering and dendrogram analysis which usually returns a heat map and a branched tree. A quick scan at these two figures provides first-hand information on variations in the samples if any. We have rephrased the statement. (Page 13; lines 320-322)

13. Lines 360, 397, and elsewhere: terms like ‘disease condition’ are vague. State the disease that you are referring to.

Response: Names of the diseases have been provided. (Page 15, line 382; page 17, line 423)

14. Line 443: Change to read “…possesses a stable personalized microbiome….”

Response: Changed as suggested (page 19’ line 471)

15. Line 460: ‘tree by’.

Response: The legend has been rephrased

16. Figure 11 panels B, C and D could be moved to the supplement since panel A is the only one that showed an interesting result.

Response: We respect the reviewer’s suggestion and have modified the fig 11. We have moved the panels B, C and D to the supplementary file section. We have added the 2D PCoA plot for better explanation of the clusters.

17. I think the authors should be careful with the use of the terms ‘longitudinal and temporal.’ I would agree that the 10 individuals that were sampled in sequential months were followed longitudinally. ‘Temporal’ can also imply the same thing as longitudinal, meaning you followed something across time. Maybe “seasonal” would be a better word since you don’t appear to sample most of the individuals more than once? This could be altered in the title.

Response: We agree with the reviewer’s recommendation and have modified the title as suggested.

---

## [Decision Letter · Decision Letter 1]

20 Sep 2021

PONE-D-21-17891R1Composition of Nasal Bacterial Community and its Seasonal Variation in Health Care Workers Stationed in a Clinical Research LaboratoryPLOS ONE

Dear Dr. Habibi,

Thank you for submitting your manuscript to PLOS ONE. After careful consideration, we feel that it has merit but does not fully meet PLOS ONE’s publication criteria as it currently stands. Therefore, we invite you to submit a revised version of the manuscript that addresses the points raised during the review process.

We look forward to receiving your revised manuscript.

Kind regards,

Christopher Staley, Ph.D.

Academic Editor

PLOS ONE

Additional Editor Comments:

In the interest of time, I am returning this without comments from a second reviewer. As noted by Reviewer 1, extensive editing for English is still required as are clarifications to the points raised by the reviewer.

Reviewers' comments:

Reviewer's Responses to Questions

**Comments to the Author**

1. If the authors have adequately addressed your comments raised in a previous round of review and you feel that this manuscript is now acceptable for publication, you may indicate that here to bypass the “Comments to the Author” section, enter your conflict of interest statement in the “Confidential to Editor” section, and submit your "Accept" recommendation.

Reviewer #2: (No Response)

2. Is the manuscript technically sound, and do the data support the conclusions?

Reviewer #2: Partly

3. Has the statistical analysis been performed appropriately and rigorously? 

Reviewer #2: I Don't Know

4. Have the authors made all data underlying the findings in their manuscript fully available?

Reviewer #2: Yes

5. Is the manuscript presented in an intelligible fashion and written in standard English?

Reviewer #2: No

6. Review Comments to the Author

Reviewer #2: The manuscript by Habibi et al., is a revised version of a manuscript that I previously reviewed. The authors have made attempts to address the prior comments, and the work is improved. However, there are still issues with the writing and some of the presented information.

1. One of my prior concerns was with the authors failure to adequately explain where all of the samples came from. To address this, they have added Table 1. This table shows the distribution of the number of samples that fell into the various epidemiologic groupings. The label over the right hand side of the data states “No of individuals sampled” and shows that these values all add up to 73 across the various breakdowns. However, the authors state in the paper that they sampled from “47” individuals. They seem to be confusing ‘samples’ and ‘individuals.’ Thus, I still have no clue how they arrived at 73 samples from 47 individuals. 10 individuals were sampled 3 times each, which would be 30 samples. If the remaining 37 individuals were sampled only once each, that would be 37 samples. 30+37=67 and not 73. Something is clearly not explained clearly or accurately. This needs to be clarified.

2. I think point #1 indicates that the work is still in need of significant language editing. The only language changes that were made appear to be in response to specific comments. Judicious editing would help with clarity and the presentation of the work; you want it to be as interpretable by the audience as possible. See a few of the specific comments below.

3. The sections added on lines 185-192 are poorly written and difficult to follow.

4. I still don’t think the authors have clearly defined in the paper what they consider to be a ‘core’ microbiome. The easiest way to address this is that the first time this phrase is use, the authors should state something along the following: ‘we defined the core microbiome as genera present in 30% of the individual samples’. If ‘genera’ is not correct, add the correct word. They can also include the reference they mention in their response to my prior comment.

5. Lines 111. Was there a set procedure or is there a published procedure for how the swab was collected? Was it inserted into a particular nostril first, twirled a certain number of times, etc.? Provide information or a reference.

6. Line 194-195. This may not be true. You have no way of knowing that this difference has anything to do with these individual’s occupations. It may simply be due to geographic differences of the sampled populations, etc.

Specifics:

1. The sentence beginning “Competition….” On line 23 seems incomplete. I’m not sure what you are trying to say.

2. It’s not clear how someone can “indirectly” work in a clinical research lab (line 25).

3. Line 38. Captalize “S” in 16S

4. Line 42. Add a comma after “ecosystem”

5. Line 53. “the key medium” should likely be “a key medium”

6. Line 54. “structure” should be “structures”

7. Line 60. The effect of what? The meaning is unclear.

8. Line 64. “disperse” should be “disperses”

9. Line 166. “picked” doesn’t seem like the best word. Perhaps “revealed a total of 2363 OTUs withing the dataset.”

10. These are only a few examples of the language issues that I’m talking about in the the major point #2 above. Virtually every paragraph has many examples like this that need to be fixed.

7. PLOS authors have the option to publish the peer review history of their article (what does this mean?). If published, this will include your full peer review and any attached files.

Reviewer #2: No

---

## [Author Response · Author response to Decision Letter 1]

1 Oct 2021

Response to reviewers

Reviewer #2: The manuscript by Habibi et al., is a revised version of a manuscript that I previously reviewed. The authors have made attempts to address the prior comments, and the work is improved. However, there are still issues with the writing and some of the presented information.

Thank you for reading our manuscript again. We are grateful for your constructive comments. This, has once again improved our manuscript. 

1. One of my prior concerns was with the authors failure to adequately explain where all of the samples came from. To address this, they have added Table 1. This table shows the distribution of the number of samples that fell into the various epidemiologic groupings. The label over the right hand side of the data states “No of individuals sampled” and shows that these values all add up to 73 across the various breakdowns. However, the authors state in the paper that they sampled from “47” individuals. They seem to be confusing ‘samples’ and ‘individuals.’ Thus, I still have no clue how they arrived at 73 samples from 47 individuals. 10 individuals were sampled 3 times each, which would be 30 samples. If the remaining 37 individuals were sampled only once each, that would be 37 samples. 30+37=67 and not 73. Something is clearly not explained clearly or accurately. This needs to be clarified.

Response: We would like to inform the reviewer that 73 samples came from 47 healthy volunteers only. From the remaining 37 individuals few were sampled more than once, making the total number of samples to 73. This information has been added to the methodology section. Lines 91-92. Also, the heading in column 4 in Table 1 has been changed from no of sample to total no. of individuals sampled in each season.

2. I think point #1 indicates that the work is still in need of significant language editing. The only language changes that were made appear to be in response to specific comments. Judicious editing would help with clarity and the presentation of the work; you want it to be as interpretable by the audience as possible. See a few of the specific comments below.

Response: We accepted the reviewers suggestion. The manuscript has now been read by a native English speaker (Ms. Christiana Cholakis, from University of Manitoba) and is hopefully free from linguistic errors.

3. The sections added on lines 185-192 are poorly written and difficult to follow.

Response: This section has been rewritten.

4. I still don’t think the authors have clearly defined in the paper what they consider to be a ‘core’ microbiome. The easiest way to address this is that the first time this phrase is use, the authors should state something along the following: ‘we defined the core microbiome as genera present in 30% of the individual samples’. If ‘genera’ is not correct, add the correct word. They can also include the reference they mention in their response to my prior comment.

Response: We have added this phrase in the methodology section with the relevant citation (Line 208-209). We have also done some changes in the text to be clearer on this aspect. For eg. Lines 106, 223, 603, 605 have been re-phrased. 

5. Lines 111. Was there a set procedure or is there a published procedure for how the swab was collected? Was it inserted into a particular nostril first, twirled a certain number of times, etc.? Provide information or a reference.

Response: A reference to the swab collection procedure has been added (Line 111)

6. Line 194-195. This may not be true. You have no way of knowing that this difference has anything to do with these individual’s occupations. It may simply be due to geographic differences of the sampled populations, etc.

Response: This sentence has been reframed and the reason of spatial variations has also been added. Line 194-195.

Specifics:

1. The sentence beginning “Competition….” On line 23 seems incomplete. I’m not sure what you are trying to say.

Response: We have modified this sentence.

2. It’s not clear how someone can “indirectly” work in a clinical research lab (line 25).

Response: We have removed the words directly and indirectly.

3. Line 38. Captalize “S” in 16S

Response: The change has been done

4. Line 42. Add a comma after “ecosystem”

Response: The requested change has been done

5. Line 53. “the key medium” should likely be “a key medium”

Response: The has been replaced by ‘a’ 

6. Line 54. “structure” should be “structures”

Response: The change has been done.

7. Line 60. The effect of what? The meaning is unclear.

Response: This sentence has been removed from the paragraph

8. Line 64. “disperse” should be “disperses”

Response: The amendment has been done

9. Line 166. “picked” doesn’t seem like the best word. Perhaps “revealed a total of 2363 OTUs withing the dataset.”

Response: We have replaced picked with the suggested phrase

10. These are only a few examples of the language issues that I’m talking about in the the major point #2 above. Virtually every paragraph has many examples like this that need to be fixed.

Response: We are once again apologetic of our ambigous expressions. The manuscript has been read by an English native speaker and is hopefully free from linguistic errors.

---

## [Decision Letter · Decision Letter 2]

20 Oct 2021

PONE-D-21-17891R2Composition of Nasal Bacterial Community and its Seasonal Variation in Health Care Workers Stationed in a Clinical Research LaboratoryPLOS ONE

Dear Dr. Habibi,

Thank you for submitting your manuscript to PLOS ONE. After careful consideration, we feel that it has merit but does not fully meet PLOS ONE’s publication criteria as it currently stands. Therefore, we invite you to submit a revised version of the manuscript that addresses the points raised during the review process. Thank you for your response to the comments from reviewer 1. A second reviewer has raised valid concerns and requested clarification on how the groups were analyzed and environmental factors were taken into account.

We look forward to receiving your revised manuscript.

Kind regards,

Christopher Staley, Ph.D.

Academic Editor

PLOS ONE

Journal Requirements:

Additional Editor Comments (if provided):

Reviewers' comments:

Reviewer's Responses to Questions

**Comments to the Author**

1. If the authors have adequately addressed your comments raised in a previous round of review and you feel that this manuscript is now acceptable for publication, you may indicate that here to bypass the “Comments to the Author” section, enter your conflict of interest statement in the “Confidential to Editor” section, and submit your "Accept" recommendation.

Reviewer #2: All comments have been addressed

Reviewer #3: (No Response)

2. Is the manuscript technically sound, and do the data support the conclusions?

Reviewer #2: Yes

Reviewer #3: Partly

3. Has the statistical analysis been performed appropriately and rigorously? 

Reviewer #2: Yes

Reviewer #3: No

4. Have the authors made all data underlying the findings in their manuscript fully available?

Reviewer #2: Yes

Reviewer #3: Yes

5. Is the manuscript presented in an intelligible fashion and written in standard English?

Reviewer #2: Yes

Reviewer #3: Yes

6. Review Comments to the Author

Reviewer #2: This is the 3rd version of this manuscript that I have reviewed. This version is much improved and I appreciate the authors efforts to address all of my prior concerns.

Reviewer #3: In this article, the authors characterized the structure and dynamics of nasal microbiome from healthy individuals in a clinical research laboratory in Kuwait. The authors also investigated factors driving the variation in nasal microbiome in this special population.

In general, it is easy to capture the main points of this paper. However, the authors need to clarify the samples they used or reconsider some statistical methods before drawing some of their conclusions.

Major issues:

1.Page 5 line 107-110, “The dataset from 88 professionals working in health care centres (HCC) in Taiwan and 20 healthy individuals (HC) working in a livestock farm in Iowa were used to compare our dataset with healthy individuals working in non-healthcare settings”. Page 8 line 195-196, “These variations were presumed due to the differences between the nasal bacterial communities of health care and non-healthcare workers.” 88 of them were working in a health center in Taiwan. I suppose they are health care workers. Why take them as part of the non-healthcare controls? Both ethnic or genetic and climate (Taiwan is tropical and Kuwait is desert climate) will be the main drivers on their nasal microbiome, especially as the authors have found the seasonal change in their samples. The other 22 samples were from a farm in the US. Exposure to the livestocks and outdoor jobs will make a great difference to their nasal microbiome. These two groups were not good controls if the authors want to demonstrate the difference between healthcare workers v.s. non-healthcare workers.

2.Page 10-12 Session “seasonal variation”. Which samples were the analysis done with? If the recruited subjects were sampled for different times, the final results will be driven by the individuals sampled more times, making the conclusions unreliable. The best practice is to conduct paired analysis for differential comparison and track the change within the subject. For the same reason, correlation between bacteria abundance across samples is not appropriate for identifying co-occurring bacteria. Consider similarity in patterns of longitudinal change instead.

3. A comparison between inter-subject dissimilarity and inter-factor dissimilarity will help the audience to understand how strong the influence of this factor is for all the analysis to investigate factors driving variation in nasal microbiome. Also, the authors have shown the instability of nasal microbiome within subjects. But, I am not sure how unstable it is. Is it still possible to tell if two samples were the same subject?

Minor issues:

1. Page 2 line 28, “The taxonomic profiling and core microbiome analysis predicted three predominant genera as Corynebacterium (15.0%), Staphylococcus (10.3%) and, Moraxella (10.0%).” Unless the authors have tested this conclusion on unseen data, “predict” is not a suitable verb here.

2. Page 2 line 45, “skin, blood, urine and, any other crevices or orifices.” Bacteria in urine come from the bladder or urinary tract, like fecal microbiome come from the gut.

3. Page 8 line 177, as multiple samples collected from the same subjects, paired statistical analysis methods will be more appropriate.

4. Page 8 line 179-180, Bray-curtis is not a good choice for compositional data dissimilarity. Besides, Bray-curtis should be between 0 to 1 and it's ratio between arithmetic sums. It cannot be used in CLR transformed data. CLR is in log scale, it's geometric and CLR data will have negative values.

5. Page 8 line 191, “Data was rarified (Fig S3) and sequences < 5 bases and quality score < 20, were filtered out.” Did the authors rarified before the quality filter?

6. Page 9 line 206-208, why is the RA of Actinomycetales higher than the RA of Actinobacteria?

7. Page 13 line 304-305, “From these observations, we concluded that an individuals’ nasal microbiome is largely defined by the prevailing season in addition to personal habits and daily routine.” Is there any evidence in this study to support the impact of “personal habits and daily routine” on nasal microbiome?

8. Page 13 line 318-329. In the alpha diversity analysis, the autumn microbiome stood out among the four seasons. But in the beta diversity, it’s quite the opposite. What can be the reason?

7. PLOS authors have the option to publish the peer review history of their article (what does this mean?). If published, this will include your full peer review and any attached files.

Reviewer #2: No

Reviewer #3: **Yes: **Nan Shen

---

## [Author Response · Author response to Decision Letter 2]

27 Oct 2021

Response to Reviewers

Reviewer #2: This is the 3rd version of this manuscript that I have reviewed. This version is much improved and I appreciate the authors efforts to address all of my prior concerns.

Response: We are very much thankful to the anonymous reviewer for his encouraging feebback.

Reviewer #3: In this article, the authors characterized the structure and dynamics of nasal microbiome from healthy individuals in a clinical research laboratory in Kuwait. The authors also investigated factors driving the variation in nasal microbiome in this special population.

In general, it is easy to capture the main points of this paper. However, the authors need to clarify the samples they used or reconsider some statistical methods before drawing some of their conclusions.

Response: We express our gratitude to the reviewer (Dr. Nan Shen) for his positive feedback. We have provided a response to his valid concerns in the following text.

Major issues:

1.Page 5 line 107-110, “The dataset from 88 professionals working in health care centres (HCC) in Taiwan and 20 healthy individuals (HC) working in a livestock farm in Iowa were used to compare our dataset with healthy individuals working in non-healthcare settings”. Page 8 line 195-196, “These variations were presumed due to the differences between the nasal bacterial communities of health care and non-healthcare workers.” 88 of them were working in a health center in Taiwan. I suppose they are health care workers. Why take them as part of the non-healthcare controls? Both ethnic or genetic and climate (Taiwan is tropical and Kuwait is desert climate) will be the main drivers on their nasal microbiome, especially as the authors have found the seasonal change in their samples. The other 22 samples were from a farm in the US. Exposure to the livestocks and outdoor jobs will make a great difference to their nasal microbiome. These two groups were not good controls if the authors want to demonstrate the difference between healthcare workers v.s. non-healthcare workers.

Response: We are in agreement with the reviewer on this aspect. As the study lacked control samples from Kuwait (Where the whole project was executed) we applied the strategy to use publicly available datasets for comparison. The studies on nasal microbiome of healthy subjects were limited, however we were able to get these two datasets. As pointed out by the reviewer and also our own understanding, Taiwan and Kuwait are two distantly apart countries and differences in the nasal microbiome of health care workers owing to the geographical differences are obvious, we are removing the word control for this dataset. 

Regarding the other dataset i.e. healthy workers (NHC) in a farm. We see the farm as a different setting than that of a health care centre. Also, these individuals were not in contact with animals on the farm and were mostly working indoors. The differences in community composition can therefore be attributed to the difference in workplace and therefore we believe it can be used as a control. We cannot ignore the environmental factors as well and this explanation has also been given in the text. (page 5 Lines 108-112; page 8 lines 195-204)

2.Page 10-12 Session “seasonal variation”. Which samples were the analysis done with? If the recruited subjects were sampled for different times, the final results will be driven by the individuals sampled more times, making the conclusions unreliable. The best practice is to conduct paired analysis for differential comparison and track the change within the subject. For the same reason, correlation between bacteria abundance across samples is not appropriate for identifying co-occurring bacteria. Consider similarity in patterns of longitudinal change instead.

Response: The analysis was performed on all the samples. Although, the individuals were sampled more than once, however were not repeated in the same season. In addition to this almost 50% of subjects sampled in a particular season were unique and therefore, we believe that the variations are due to seasonality. We would also like to bring to the reviewer’s attention that pairwise comparisons on alpha diversities were already done through ANOVA an analogue of paired t-test. However, an additional comparison between summer vs winter; summer vs spring; summer vs autumn; winter vs spring; winter vs autumn and spring vs autumn has been performed. The differential abundance analysis on non-parametric Wilcoxson rank sum test at a confidence interval of p<0.05 was used for this. This compared the median abundances within a particular season. The results have been added in the manuscript (Page 11 Lines 268-285 and new fig 6). 

3. A comparison between inter-subject dissimilarity and inter-factor dissimilarity will help the audience to understand how strong the influence of this factor is for all the analysis to investigate factors driving variation in nasal microbiome. Also, the authors have shown the instability of nasal microbiome within subjects. But, I am not sure how unstable it is. Is it still possible to tell if two samples were the same subject?

Response: Thank you for bringing this to our notice. We have done the pairwise comparisons employing a student’s t-test on two variables that are inter-subject dissimilarity and inter-factor dissimilarity. All the results were highly significant p-value < 0.05. The variance table has been provided in the manuscript. The nasal microbiome differed significantly as proven by the t-test. (page 13 lines 328-331; table 4)

Taking apart the prevalent genera, a good number of other genera were found in each subject when sampled at a different time point. Therefore, we assume that the microbiome might not be same for the same subject if sampled at a different time point. 

Minor issues:

1. Page 2 line 28, “The taxonomic profiling and core microbiome analysis predicted three predominant genera as Corynebacterium (15.0%), Staphylococcus (10.3%) and, Moraxella (10.0%).” Unless the authors have tested this conclusion on unseen data, “predict” is not a suitable verb here.

Response: This word has been replaced by identified. (page 2Line 28)

2. Page 2 line 45, “skin, blood, urine and, any other crevices or orifices.” Bacteria in urine come from the bladder or urinary tract, like fecal microbiome come from the gut.

Response: blood and urine have been removed from this sentence (Page 2 line 44-45)

3. Page 8 line 177, as multiple samples collected from the same subjects, paired statistical analysis methods will be more appropriate.

Response: Please see the response 2 and 3 under major issues.

4. Page 8 line 179-180, Bray-curtis is not a good choice for compositional data dissimilarity. Besides, Bray-curtis should be between 0 to 1 and it's ratio between arithmetic sums. It cannot be used in CLR transformed data. CLR is in log scale, it's geometric and CLR data will have negative values.

Response: We have opted for the Bray Curtis dissimilarity based on several studies employing this method. We still prefer to use it. We have now used the TSS method for data normalization. The values and the figures have been replaced accordingly. (Page 6; line 183; page 14 line 353-354, fig 13)

5. Page 8 line 191, “Data was rarified (Fig S3) and sequences < 5 bases and quality score < 20, were filtered out.” Did the authors rarified before the quality filter?

Response: Data was first processed for quality check and then rarified. We have done this change in the text. (page 8, line 194)

6. Page 9 line 206-208, why is the RA of Actinomycetales higher than the RA of Actinobacteria?

Response: This was a typo error and has been resolved. Page 9 line 215

7. Page 13 line 304-305, “From these observations, we concluded that an individuals’ nasal microbiome is largely defined by the prevailing season in addition to personal habits and daily routine.” Is there any evidence in this study to support the impact of “personal habits and daily routine” on nasal microbiome?

Response: we have replaced the word concluded with assumed. Page 14; line 332

8. Page 13 line 318-329. In the alpha diversity analysis, the autumn microbiome stood out among the four seasons. But in the beta diversity, it’s quite the opposite. What can be the reason?

Response: Alpha diversity measures within the group diversity, whereas beta diversity is measuring among the group diversity. In autumn season the species richness, evenness and abundances might be high whereas these population might be overlapping with other seasons. This is the only reason we think are responsible for the differences in alpha and beta diversities of autumn season.

---

## [Decision Letter · Decision Letter 3]

8 Nov 2021

Composition of Nasal Bacterial Community and its Seasonal Variation in Health Care Workers Stationed in a Clinical Research Laboratory

PONE-D-21-17891R3

Dear Dr. Habibi,

We’re pleased to inform you that your manuscript has been judged scientifically suitable for publication and will be formally accepted for publication once it meets all outstanding technical requirements.

Kind regards,

Christopher Staley, Ph.D.

Academic Editor

PLOS ONE

Additional Editor Comments (optional):

Reviewers' comments:

Reviewer's Responses to Questions

**Comments to the Author**

1. If the authors have adequately addressed your comments raised in a previous round of review and you feel that this manuscript is now acceptable for publication, you may indicate that here to bypass the “Comments to the Author” section, enter your conflict of interest statement in the “Confidential to Editor” section, and submit your "Accept" recommendation.

Reviewer #3: All comments have been addressed

2. Is the manuscript technically sound, and do the data support the conclusions?

Reviewer #3: Yes

3. Has the statistical analysis been performed appropriately and rigorously? 

Reviewer #3: Yes

4. Have the authors made all data underlying the findings in their manuscript fully available?

Reviewer #3: Yes

5. Is the manuscript presented in an intelligible fashion and written in standard English?

Reviewer #3: Yes

6. Review Comments to the Author

Reviewer #3: All the comments have been well addressed by the authors. Only one suggest is that in Page 11 line 268-279, change "(0.000)" to "p < 0.001" or scientific notation.

7. PLOS authors have the option to publish the peer review history of their article (what does this mean?). If published, this will include your full peer review and any attached files.

Reviewer #3: No

---

## [Editor Report · Acceptance letter]

12 Nov 2021

PONE-D-21-17891R3 

Composition of Nasal Bacterial Community and its Seasonal Variation in Health Care Workers Stationed in a Clinical Research Laboratory 

Dear Dr. Habibi:

I'm pleased to inform you that your manuscript has been deemed suitable for publication in PLOS ONE. Congratulations! Your manuscript is now with our production department. 

Kind regards, 

on behalf of

Dr. Christopher Staley 

Academic Editor

PLOS ONE